# On the Dynamics and Convergence of Weight Normalization for Training Neural Networks

## Abstract

We present a proof of convergence for ReLU networks trained with weight normalization. In the analysis, we consider over-parameterized 2-layer ReLU networks initialized at random and trained with batch gradient descent and a fixed step size. The proof builds on recent theoretical works that bound the trajectory of parameters from their initialization and monitor the network predictions via the evolution of a "neural tangent kernel" (Jacot et al. 2018). We discover that training with weight normalization decomposes such a kernel via the so called "length-direction decoupling". This in turn leads to two convergence regimes and can rigorously explain the utility of WeightNorm. From the modified convergence we make a few curious observations including a natural form of "lazy training" where the direction of each weight vector remains stationary.

## 1 Introduction

Dynamic normalization in neural networks is a re-parametrization procedure between the layers that improves stability during training and leads to faster convergence. This approach was popularized with the introduction of Batch Normalization (BatchNorm) in [20] and has led to a plethora of additional normalization methods, notably including Layer Normalization (LayerNorm) [6] and Weight Normalization (WeightNorm) [28]. WeightNorm was proposed as a method that emulates BatchNorm and benefits from similar stability and convergence properties. Moreover, WeightNorm has the advantage of not requiring a batch setting, therefore considerably reducing the computational overhead that is imposed by BatchNorm [16]. WeightNorm is widely used in training of neural networks and is the focus of this work.

Today, normalization methods are ubiquitous in the training of neural nets since in practice they significantly improve the convergence speed and stability in training. Despite the impressive empirical results and massive popularity of dynamic normalization methods, explaining their utility and proving that they converge when training with non-smooth, non-convex loss functions has remained an unsolved problem. In this paper we provide sufficient conditions on the data, initialization, and over-parameterization for dynamically normalized ReLU networks to converge to a global minimum of the loss function, and rigorously illustrate the utility of normalization methods.

Consider the class of 2-layer ReLU neural networks $f : \mathbb{R}^d \to \mathbb{R}$ parameterized by $(\mathbf{W}, \mathbf{c}) \in \mathbb{R}^{m \times d} \times \mathbb{R}^m$ as

$$f(\mathbf{x}; \mathbf{W}, \mathbf{c}) = \frac{1}{\sqrt{m}} \sum_{k=1}^{m} c_k \sigma(\mathbf{w}_k^\top \mathbf{x}). \tag{1.1}$$

Here the activation function is the ReLU, $\sigma(s) = \max\{s, 0\}$ [26], $m$ denotes the width of the second layer, and $f$ is normalized accordingly by a factor $\sqrt{m}$. We investigate gradient descent training with WeightNorm for (1.1), which re-parameterizes the network in terms of $(\mathbf{V}, \mathbf{g}, \mathbf{c}) \in \mathbb{R}^{m \times d} \times \mathbb{R}^m \times \mathbb{R}^m$ as

$$f(\mathbf{x}; \mathbf{V}, \mathbf{g}, \mathbf{c}) = \frac{1}{\sqrt{m}} \sum_{k=1}^{m} c_k \sigma\left( g_k \cdot \frac{\mathbf{v}_k^\top \mathbf{x}}{\|\mathbf{v}_k\|_2} \right). \tag{1.2}$$

This gives a similar parameterization to [14] that study convergence of gradient optimization of convolutional filters on Gaussian data. We consider the regression task, optimizing with respect to the

$L^2$ loss with random parameter initialization and focus on the over-parametrized regime, meaning that $m > n$, where $n$ is the number of training samples.

The neural network function class (1.1) has been studied in many papers including [3, 15, 33, 36] along with other similar over-parameterized architectures [1, 14, 23]. An exuberant series of recent works prove that feed-forward ReLU networks converge to zero training error when trained with gradient descent from random initialization. Nonetheless, to the best of our knowledge, there are no proofs that ReLU networks trained with *dynamic normalization* on general data converge to a global minimum. This is in part because normalization methods completely change the optimization landscape during training. Here we show that neural networks of the form given above converge at linear rate when trained with gradient descent and WeightNorm. The analysis is based on the over-parameterization of the networks, which allows for guaranteed descent while the gradient is non-zero.

For regression training, a group of papers studied the trajectory of the networks' predictions and showed that they evolve via a "neural tangent kernel" (NTK) as introduced by Jacot et al. [21]. The latter paper studies neural network convergence in the continuous limit of infinite width over-parameterization, while the works of [3, 15, 27, 33, 36] analyze the finite width setting. For finite-width over-parameterized networks, the training evolution also exhibits a kernel that takes the form of a Gram matrix. In these works, the convergence rate is dictated by the least eigenvalue of the kernel. We build on this fact, and also on the general ideas of the proof of [15] and the refined work of [3].

Compared with un-normalized training, we prove that normalized networks follow a modified kernel evolution that features a "length-direction" decomposition of the NTK. This leads to two convergence regimes in WeightNorm training and explains the utility of WeightNorm from the perspective of the NTK. In the settings considered, WeightNorm significantly reduces the amount of over-parameterization needed for provable convergence, as compared with un-normalized settings. The decomposition of the NTK also connects to observations of [12] that discuss "lazy training" which refers to a training regime where the weights of the network stay close to their initialization (see Section 6). Further, we present a more careful analysis that leads to improved over-parameterization bounds as compared with [15].

In this work we rigorously analyze the dynamics of weight normalization training and its convergence from the perspective of the neural tangent kernel. We discover WeightNorm training has two regimes with distinct behaviors. The main contributions of this work are:

- We prove the first general convergence result for *dynamically normalized* 2-layer ReLU networks trained with gradient descent. Our formulation does not assume the existence of a teacher network and has mild assumptions on the training data.

- We explain the utility of normalization methods via a decomposition of the neural tangent kernel. In the analysis we highlight two distinct convergence regimes and give a concrete example of "lazy training" for finite-step gradient descent.

- It is shown that finite-step gradient descent converges for all weight magnitudes at initialization and we significantly reduce the amount of over-parameterization required for provable convergence as compared with un-normalized training.

The paper is organized as follows. In Section 2 we provide background on WeightNorm and derive key evolution dynamics of training in Section 3. We present and discuss our main results, alongside with the idea of the proof, in Section 4. We discuss related work in Section 5, and offer a discussion of our results and their implications to dynamic normalization training and "lazy training" in Section 6. Proofs are presented in the Appendix.

## 2 WEIGHTNORM

Here we give an overview of the WeightNorm procedure and review some known properties of normalization methods.

**Notation** We use lowercase, lowercase boldface, and uppercase boldface letters to denote scalars, vectors and matrices resp. We denote the Rademacher distribution as $U\{1, -1\}$ and write $N(\boldsymbol{\mu}, \boldsymbol{\Sigma})$ for a Gaussian with mean $\boldsymbol{\mu}$ and covariance $\boldsymbol{\Sigma}$. Training points are denoted by $\mathbf{x}_1 \ldots \mathbf{x}_n \in \mathbb{R}^d$ and parameters of the first layer by $\mathbf{v}_k \in \mathbb{R}^d$. We use $\sigma(x) := \max\{x, 0\}$, and write $\|\cdot\|_2, \|\cdot\|_F$ for

the spectral and Frobenius norms for matrices. $\lambda_{\min}(\mathbf{A})$ is used to denote the minimum eigenvalue of a matrix $\mathbf{A}$ and $\langle \cdot, \cdot \rangle$ denotes the Euclidean inner product. For a vector $\mathbf{v}$ denote the $\ell_2$ vector norm as $\|\mathbf{v}\|_2$ and for a positive definite matrix $\mathbf{S}$ define the induced vector norm $\|\mathbf{v}\|_{\mathbf{S}} := \sqrt{\mathbf{v}^\top \mathbf{S} \mathbf{v}}$. The projections of $\mathbf{x}$ onto $\mathbf{u}$ and $\mathbf{u}^\perp$ are defined as $\mathbf{x}^{\mathbf{u}} := \frac{\mathbf{u}\mathbf{u}^\top \mathbf{x}}{\|\mathbf{u}\|_2^2}$, $\mathbf{x}^{\mathbf{u}^\perp} := \mathbf{x}\big(\mathbf{I} - \frac{\mathbf{u}\mathbf{u}^\top}{\|\mathbf{u}\|_2^2}\big)$. Denote the indicator function of event $A$ as $\mathbb{1}_A$ and for a weight vector at time $t$, $\mathbf{v}_k(t)$, and data point $\mathbf{x}_i$ we denote $\mathbb{1}_{ik}(t) := \mathbb{1}_{\{\mathbf{v}_k(t)^\top \mathbf{x}_i \geq 0\}}$.

**WeightNorm procedure** For a single neuron $\sigma(\mathbf{w}^\top \mathbf{x})$, WeightNorm re-parametrizes the weight $\mathbf{w} \in \mathbb{R}^d$ in terms of $\mathbf{v} \in \mathbb{R}^d$, $g \in \mathbb{R}$ as

$$\mathbf{w}(\mathbf{v}, g) = g \cdot \frac{\mathbf{v}}{\|\mathbf{v}\|_2}, \quad \sigma\left(g \cdot \frac{\mathbf{v}^\top \mathbf{x}}{\|\mathbf{v}\|_2}\right). \tag{2.1}$$

This decouples the magnitude and direction of each weight vector (referred as the "length-direction" decomposition). In comparison, for BatchNorm each output $\mathbf{w}^\top \mathbf{x}$ is normalized according to the average statistics in a batch. We can draw the following analogy between WeightNorm and BatchNorm if the inputs $\mathbf{x}_i$ are centered ($\mathbb{E}\mathbf{x} = \mathbf{0}$) and the covariance matrix is known ($\mathbb{E}\mathbf{x}\mathbf{x}^\top = \mathbf{S}$). In this case, batch training with BatchNorm amounts to

$$\sigma\left(\gamma \cdot \frac{\mathbf{w}^\top \mathbf{x}}{\sqrt{\mathbb{E}_{\mathbf{x}}(\mathbf{w}^\top \mathbf{x}\mathbf{x}^\top \mathbf{w})}}\right) = \sigma\left(\gamma \cdot \frac{\mathbf{w}^\top \mathbf{x}}{\sqrt{\mathbf{w}^\top \mathbf{S}\mathbf{w}}}\right) = \sigma\left(\gamma \cdot \frac{\mathbf{w}^\top \mathbf{x}}{\|\mathbf{w}\|_{\mathbf{S}}}\right). \tag{2.2}$$

From this prospective, WeightNorm is a special case of (2.2) with $\mathbf{S} = \mathbf{I}$ [22, 28].

**Properties of WeightNorm** We start by giving an overview of known properties of WeightNorm that will be used to derive the gradient flow dynamics of WeightNorm training.

For re-parameterization (2.1) of a network function $f$ that is initially parameterized with a weight $\mathbf{w}$, the gradient $\nabla_\mathbf{w} f$ relates to the gradients $\nabla_\mathbf{v} f$, $\nabla_g f$ by the identities

$$\nabla_\mathbf{v} f = \frac{g}{\|\mathbf{v}\|_2}(\nabla_\mathbf{w} f)^{\mathbf{v}^\perp}, \quad \nabla_g f = (\nabla_\mathbf{w} f)^{\mathbf{v}}.$$

This implies that $\nabla_\mathbf{v} f \cdot \mathbf{v} = 0$ for each input $\mathbf{x}$ and parameter $\mathbf{v}$. For gradient flow, this orthogonality results in $\|\mathbf{v}(0)\|_2 = \|\mathbf{v}(t)\|_2$ for all $t$. For gradient descent (with step size $\eta$) the discretization in conjunction with orthogonality leads to increasing parameter magnitudes during training [4, 19, 28], as illustrated in Figure 1,

$$\|\mathbf{v}(s+1)\|_2^2 = \|\mathbf{v}(s)\|_2^2 + \eta^2 \|\nabla_\mathbf{v} f\|_2^2 \geq \|\mathbf{v}(s)\|_2^2. \tag{2.3}$$

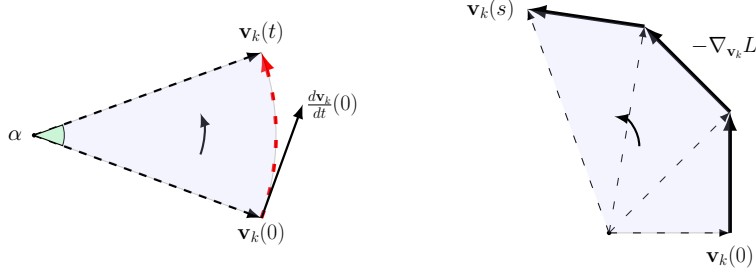

Figure 1: WeightNorm updates for gradient flow and gradient descent. For gradient flow, the norm of the weights are preserved, i.e., $\|\mathbf{v}_k(0)\|_2 = \|\mathbf{v}_k(t)\|_2$ for all $t > 0$. For gradient descent, the norm of the weights $\|\mathbf{v}_k(s)\|_2$ is increasing with $s$.

**Problem Setup** We analyze (1.1) with WeightNorm training (1.2), so that

$$f(\mathbf{x}; \mathbf{V}, \mathbf{c}, \mathbf{g}) = \frac{1}{\sqrt{m}} \sum_{k=1}^m c_k \sigma\left(g_k \cdot \frac{\mathbf{v}_k^\top \mathbf{x}}{\|\mathbf{v}_k\|_2}\right).$$

We take an initialization in the spirit of [17, 28]:

$$\mathbf{v}_k(0) \sim N(0, \alpha^2\mathbf{I}), \quad c_k \sim U\{-1, 1\}, \quad \text{and} \quad g_k(0) = \|\mathbf{v}_k(0)\|_2/\alpha. \tag{2.4}$$

Where $\alpha$ is the variance of $\mathbf{v}_k$ at initialization. The initialization of $g_k(0)$ is therefore taken to be independent of $\alpha$. We remark that the initialization (2.4) gives the same initial output distribution as in methods that study the un-normalized network class (1.1). The parameters of the network are optimized using the training data $\{(\mathbf{x}_1, y_1), \ldots, (\mathbf{x}_n, y_n)\}$ with respect to the square loss

$$L(f) = \frac{1}{2}\sum_{i=1}^{n}(f(\mathbf{x}_i) - y_i)^2 = \frac{1}{2}\|\mathbf{f} - \mathbf{y}\|_2^2, \tag{2.5}$$

where $\mathbf{f} = (f_1, f_2, \ldots, f_n)^\top = (f(\mathbf{x}_1), f(\mathbf{x}_2), \ldots, f(\mathbf{x}_n))^\top$ and $\mathbf{y} = (y_1, y_2, \ldots, y_n)^\top$.

## 3 EVOLUTION DYNAMICS

We present the gradient flow dynamics of training (2.5) to illuminate the modified dynamics of WeightNorm as compared with vanilla gradient descent. In Appendix C we tackle gradient descent training with WeightNorm where the predictions' evolution vector $\frac{d\mathbf{f}}{dt}$ is replaced by the finite difference $\mathbf{f}(s + 1) - \mathbf{f}(s)$. For gradient flow, each parameter is updated in the negative direction of the partial derivative of the loss with respect to that parameter. The optimization dynamics give

$$\frac{d\mathbf{v}_k}{dt} = -\frac{\partial L}{\partial \mathbf{v}_k}, \quad \frac{dg_k}{dt} = -\frac{\partial L}{\partial g_k}. \tag{3.1}$$

We consider the case where we fix the top layer parameters $c_k$ during training. In the over-parameterized regime, the dynamics of $c_k$ and $g_k$ turn out to be the same.

To quantify convergence, we monitor the time derivative of the $i$-th prediction, which is computed via the chain rule as

$$\frac{\partial f_i}{\partial t} = \sum_{k=1}^{m}\frac{\partial f_i}{\partial \mathbf{v}_k}\frac{d\mathbf{v}_k}{dt} + \frac{\partial f_i}{\partial g_k}\frac{dg_k}{dt}.$$

Substituting (3.1) into the $i$-th prediction evolution and grouping terms yields

$$\frac{\partial f_i}{\partial t} = -\underbrace{\sum_{k=1}^{m}\frac{\partial f_i}{\partial \mathbf{v}_k}\frac{\partial L}{\partial \mathbf{v}_k}}_{T_\mathbf{v}^i} - \underbrace{\sum_{k=1}^{m}\frac{\partial f_i}{\partial g_k}\frac{\partial L}{\partial g_k}}_{T_g^i}. \tag{3.2}$$

The gradients of $f_i$ and $L$ with respect to $\mathbf{v}_k$ are written explicitly as

$$\frac{\partial f_i}{\partial \mathbf{v}_k}(t) = \frac{1}{\sqrt{m}}\frac{c_k \cdot g_k(t)}{\|\mathbf{v}_k(t)\|_2}\cdot\mathbf{x}_i^{\mathbf{v}_k(t)^\perp}\mathbb{1}_{ik}(t), \quad \frac{\partial L}{\partial \mathbf{v}_k}(t) = \frac{1}{\sqrt{m}}\sum_{i=1}^{n}(f_i(t) - y_i)\frac{c_k \cdot g_k(t)}{\|\mathbf{v}_k(t)\|_2}\mathbf{x}_i^{\mathbf{v}_k(t)^\perp}\mathbb{1}_{ik}(t).$$

Thus $T_\mathbf{v}^i(t)$ in (3.2) can be calculated as

$$T_\mathbf{v}^i(t) = \sum_{j=1}^{n}(f_j(t) - y_j)\frac{1}{m}\sum_{k=1}^{m}\mathbb{1}_{jk}(t)\mathbb{1}_{ik}(t)\left(\frac{c_k \cdot g_k(t)}{\|\mathbf{v}_k(t)\|_2}\right)^2\langle\mathbf{x}_j^{\mathbf{v}_k(t)^\perp}, \mathbf{x}_i^{\mathbf{v}_k(t)^\perp}\rangle.$$

Defining the $\mathbf{v}$-orthogonal Gram matrix $\mathbf{V}(t)$ as

$$\mathbf{V}_{ij}(t) = \frac{1}{m}\sum_{k=1}^{m}\left(\frac{\alpha c_k \cdot g_k(t)}{\|\mathbf{v}_k(t)\|_2}\right)^2\langle\mathbf{x}_i^{\mathbf{v}_k(t)^\perp}, \mathbf{x}_j^{\mathbf{v}_k(t)^\perp}\rangle\mathbb{1}_{ik}(t)\mathbb{1}_{jk}(t), \tag{3.3}$$

we can write $T_\mathbf{v}^i$ as

$$T_\mathbf{v}^i(t) = \sum_{j=1}^{n}\frac{\mathbf{V}_{ij}(t)}{\alpha^2}(f_j(t) - y_j).$$

Note that $\mathbf{V}(t)$ is the induced neural tangent kernel [21] for the parameters $\mathbf{v}$ of WeightNorm training. While it resembles the Gram matrix $\mathbf{H}(t)$ studied in [3], here we obtain a matrix that is not piece-wise

constant in $\mathbf{v}$ since the data points are projected onto the orthogonal component of $\mathbf{v}$. We compute $T_{\mathbf{g}}^i$ in (3.2) analogously. The associated derivatives with respect to $g_k$ are

$$\frac{\partial f_i}{\partial g_k}(t) = \frac{1}{\sqrt{m}} \frac{c_k}{\|\mathbf{v}_k(t)\|_2} \sigma(\mathbf{v}_k(t)^\top \mathbf{x}_i), \quad \frac{\partial L}{\partial g_k}(t) = \frac{1}{\sqrt{m}} \sum_{j=1}^n (f_j(t) - y_j) \frac{c_k}{\|\mathbf{v}_k(t)\|_2} \sigma(\mathbf{v}_k(t)^\top \mathbf{x}_j),$$

and we obtain

$$T_{\mathbf{g}}^i(t) = \sum_{k=1}^m \frac{1}{m} \sum_{j=1}^n (f_j(t) - y_j) \left( \frac{c_k}{\|\mathbf{v}_k(t)\|_2} \right)^2 \sigma(\mathbf{v}_k(t)^\top \mathbf{x}_j) \sigma(\mathbf{v}_k(t)^\top \mathbf{x}_i).$$

Given that $c_k^2 = 1$, define $\mathbf{G}(t)$ as

$$\mathbf{G}_{ij}(t) = \frac{1}{m} \sum_{k=1}^m \frac{\sigma(\mathbf{v}_k(t)^\top \mathbf{x}_i) \sigma(\mathbf{v}_k(t)^\top \mathbf{x}_j)}{\|\mathbf{v}_k(t)\|_2^2} \tag{3.4}$$

hence we can write

$$T_{\mathbf{g}}^i(t) = \sum_{j=1}^n \mathbf{G}_{ij}(t)(f_j(t) - y_j).$$

Combining $T_{\mathbf{v}}$ and $T_{\mathbf{g}}$, the full evolution dynamics are given by

$$\frac{d\mathbf{f}}{dt} = -\left( \frac{\mathbf{V}(t)}{\alpha^2} + \mathbf{G}(t) \right)(\mathbf{f}(t) - \mathbf{y}). \tag{3.5}$$

Denote $\mathbf{\Lambda}(t) := \frac{\mathbf{V}(t)}{\alpha^2} + \mathbf{G}(t)$ and write $\frac{d\mathbf{f}}{dt} = -\mathbf{\Lambda}(t)(\mathbf{f}(t) - \mathbf{y})$. We note that $\mathbf{V}(0), \mathbf{G}(0)$, defined in (3.3), (3.4), are independent of $\alpha$:

**Observation 1** ($\alpha$ independence). *For initialization* (2.4) *and* $\alpha > 0$ *the Gram matrices* $\mathbf{V}(0), \mathbf{G}(0)$ *are independent of* $\alpha$.

This fact is proved in Appendix A. When training the neural network in (1.1) without WeightNorm (see [3, 15, 36]), the corresponding neural tangent kernel $\mathbf{H}(t)$ is defined by $\frac{\partial f_i}{\partial t} = \sum_{k=1}^m \frac{\partial f_i}{\partial \mathbf{w}_k} \frac{d\mathbf{w}_k}{dt} = -\sum_{k=1}^m \frac{\partial f_i}{\partial \mathbf{w}_k} \frac{\partial L}{\partial \mathbf{w}_k} = -\sum_{j=1}^n \mathbf{H}_{ij}(t)(f_j - y_j)$ and takes the form

$$\mathbf{H}_{ij}(t) = \frac{1}{m} \sum_{k=1}^m \mathbf{x}_i^\top \mathbf{x}_j \mathbb{1}_{ik}(t) \mathbb{1}_{jk}(t). \tag{3.6}$$

The analysis presented above shows that vanilla and WeightNorm gradient descent are related as follows.

**Proposition 1.** *Define* $\mathbf{V}(0)$, $\mathbf{G}(0)$, *and* $\mathbf{H}(0)$ *as in* (3.3), (3.4), *and* (3.6) *respectively. then for all* $\alpha > 0$,

$$\mathbf{V}(0) + \mathbf{G}(0) = \mathbf{H}(0).$$

*Thus, for* $\alpha = 1$,

$$\frac{\partial \mathbf{f}}{\partial t} = -\mathbf{\Lambda}(0)(\mathbf{f}(0) - \mathbf{y}) = -\mathbf{H}(0)(\mathbf{f}(0) - \mathbf{y}).$$

That is, WeightNorm decomposes the NTK in each layer into a length and a direction component. We refer to this as the "length-direction decoupling" of the NTK, in analogy to (2.1). From the proposition, normalized and un-normalized training kernels initially coincide if $\alpha = 1$. The utility of normalization methods can be attributed to the modified NTK $\mathbf{\Lambda}(t)$ that occurs when the WeightNorm coefficient, $\alpha$, deviates from 1. For $\alpha \gg 1$ the kernel $\mathbf{\Lambda}(t)$ is dominated by $\mathbf{G}(t)$, and for $\alpha \ll 1$ the kernel $\mathbf{\Lambda}(t)$ is dominated by $\mathbf{V}(t)$. We elaborate onthe details of this in the next section. In practice, by (2.3) as training progresses, $\|\mathbf{v}_k\|_2$ grow monotonically, leading to larger $\alpha$ and a transition from the $\mathbf{V}$-dominated to the $\mathbf{G}$-dominated regime. In our analysis we will study the two regimes $\alpha > 1$ and $\alpha < 1$ in turn.

# 4 MAIN CONVERGENCE THEORY

In this section we discuss our convergence theory and main results. From the continuous flow (3.5), we observe that the convergence behavior is described by $\mathbf{V}(t)$ and $\mathbf{G}(t)$. The matrices $\mathbf{V}(t)$ and $\mathbf{G}(t)$ are positive semi-definite since they can be shown to be covariance matrices. This implies that the least eigenvalue of the evolution matrix $\mathbf{\Lambda}(t) = \frac{1}{\alpha^2}\mathbf{V}(t) + \mathbf{G}(t)$ is bounded below by the least eigenvalue of each kernel matrix,

$$\lambda_{\min}(\mathbf{\Lambda}(t)) \geq \max\{\lambda_{\min}(\mathbf{V}(t))/\alpha^2, \lambda_{\min}(\mathbf{G}(t))\}.$$

For finite-step gradient descent, a discrete analog of evolution (3.5) holds. However, the discrete case requires additional care in ensuring dominance of the driving gradient terms. For gradient flow, it is relatively easy to see linear convergence is attained by relating the rate of change of the loss to the magnitude of the loss. Suppose that for all $t \geq 0$,

$$\lambda_{\min}\big(\mathbf{\Lambda}(t)\big) \geq \omega/2, \quad \text{with } \omega > 0. \tag{4.1}$$

Then the change in the regression loss is written as

$$\frac{d}{dt}\|\mathbf{f}(t) - \mathbf{y}\|_2^2 = 2(\mathbf{f}(t) - \mathbf{y})^\top \frac{d\mathbf{f}(t)}{dt} = -2(\mathbf{f}(t) - \mathbf{y})^\top \mathbf{\Lambda}(t)(\mathbf{f}(t) - \mathbf{y})$$

$$\overset{(4.1)}{\leq} -\omega\|\mathbf{f}(t) - \mathbf{y}\|_2^2.$$

Integrating this time derivative and using the initial conditions yields

$$\|\mathbf{f}(t) - \mathbf{y}\|_2^2 \leq \exp(-\omega t)\|\mathbf{f}(0) - \mathbf{y}\|_2^2,$$

which gives linear convergence. The focus of our proof is therefore showing that (4.1) holds throughout training.

By Observation 1 we have that $\mathbf{V}$ and $\mathbf{G}$ are independent of the WeightNorm coefficient $\alpha$ ($\alpha$ only appears in the $1/\alpha^2$ scaling of $\mathbf{\Lambda}$). This suggests that the kernel $\mathbf{\Lambda}(t) = \frac{1}{\alpha^2}\mathbf{V}(t) + \mathbf{G}(t)$ can be split into two regimes: When $\alpha < 1$ the kernel is dominated by the first term $\frac{1}{\alpha^2}\mathbf{V}$, and when $\alpha > 1$ the kernel is dominated by the second term $\mathbf{G}$. We divide our convergence result based on these two regimes.

In each regime, (4.1) holds if the corresponding dominant kernel, $\mathbf{V}(t)$ or $\mathbf{G}(t)$, maintains a positive least eigenvalue. Having a least eigenvalue that is bounded from 0 gives a convex-like property that allows us to prove convergence. To ensure that condition (4.1) is satisfied, for each regime we show that the corresponding dominant kernel is "anchored" (remains close) to an auxiliary Gram matrix which we define in the following for $\mathbf{V}$ and $\mathbf{G}$.

Define the auxiliary $\mathbf{v}$-orthogonal and $\mathbf{v}$-aligned Gram matrices $\mathbf{V}^\infty, \mathbf{G}^\infty$ as

$$\mathbf{V}_{ij}^\infty := \mathbb{E}_{\mathbf{v} \sim N(0,\alpha^2\mathbf{I})} \langle \mathbf{x}_i^{\mathbf{v}^\perp}, \mathbf{x}_j^{\mathbf{v}^\perp} \rangle \mathbb{1}_{ik}(0)\mathbb{1}_{jk}(0), \tag{4.2}$$

$$\mathbf{G}_{ij}^\infty := \mathbb{E}_{\mathbf{v} \sim N(0,\alpha^2\mathbf{I})} \langle \mathbf{x}_i^{\mathbf{v}}, \mathbf{x}_j^{\mathbf{v}} \rangle \mathbb{1}_{ik}(0)\mathbb{1}_{jk}(0). \tag{4.3}$$

For now, assume that $\mathbf{V}^\infty$ and $\mathbf{G}^\infty$ are positive definite with a least eigenvalue bounded below by $\omega$ (we give a proof sketch below). In the convergence proof we will utilize over-parameterization to ensure that $\mathbf{V}(t), \mathbf{G}(t)$ concentrate to their auxiliary versions so that they are also positive definite with a least eigenvalue that is greater than $\omega/2$. The precise formulations are presented in Lemmas B.4 and B.5 that are relegated to Appendix B.

To prove our convergence results we make the assumption that the $\mathbf{x}_i$s have bounded norm and are not parallel.

**Assumption 1** (Normalized non-parallel data)**.** *The data points* $(\mathbf{x}_1, y_1), \ldots, (\mathbf{x}_n, y_n)$ *satisfy* $\|\mathbf{x}_i\|_2 \leq 1$ *and for each index pair* $i \neq j$, $\mathbf{x}_i \neq \beta \cdot \mathbf{x}_j$ *for all* $\beta \in \mathbb{R} \setminus \{0\}$.

In order to simplify the presentation of our results, we assume that the input dimension $d$ is not too small, whereby $d \geq 50$ suffices. This is not essential for the proof. Specific details are provided in Appendix A.

**Assumption 2.** *For data points* $\mathbf{x}_i \in \mathbb{R}^d$ *assume that* $d \geq 50$.

Both assumptions can be easily satisfied by pre-processing, e.g., normalizing and shifting the data, and adding zero coordinates if needed.

Given Assumption 1, $\mathbf{V}^\infty, \mathbf{G}^\infty$ are shown to be positive definite.

**Lemma 4.1.** *Fix training data points $\{(\mathbf{x}_1, y_1), \ldots, (\mathbf{x}_n, y_n)\}$ satisfying Assumption 1. Then the $\mathbf{v}$-orthogonal and $\mathbf{v}$-aligned Gram matrices $\mathbf{V}^\infty$ and $\mathbf{G}^\infty$, defined as in (4.2) and (4.3), are strictly positive definite. We denote the least eigenvalues $\lambda_{\min}(\mathbf{V}^\infty) =: \lambda_0$, $\lambda_{\min}(\mathbf{G}^\infty) =: \mu_0$.*

**Proof sketch**  Here we sketch the proof of Lemma 4.1. The main idea, is the same as [15], is to regard the auxiliary matrices $\mathbf{V}^\infty, \mathbf{G}^\infty$ as the covariance matrices of linearly independent operators. For each data point $\mathbf{x}_i$, define $\phi_i(\mathbf{v}) := \mathbf{x}_i^{\mathbf{v}^\perp} \mathbb{1}_{\{\mathbf{x}_i^\top \mathbf{v} \geq 0\}}$. The Gram matrix $\mathbf{V}^\infty$ is the covariance matrix of $\{\phi_i\}_{i=1:n}$ taken over $\mathbb{R}^d$ with the measure $N(0, \alpha^2 \mathbf{I})$. Hence showing that $\mathbf{V}^\infty$ is strictly positive definite is equivalent to showing that $\{\phi_i\}_{i=1,\ldots n}$ are linearly independent. Unlike [15], the functionals under consideration are not piecewise constant so a different construction is used to prove independence. Analogously, a new set of operators, $\theta_i(\mathbf{v}) := \sigma(\mathbf{x}_i^{\mathbf{v}})$, is constructed for $\mathbf{G}^\infty$. Interestingly, each $\phi_i$ corresponds to $\frac{d\theta_i}{d\mathbf{v}}$. The full proof is presented in Appendix D. As already observed from evolution (3.5), different magnitudes of $\alpha$ lead to two distinct regimes that are discussed below. We present the main results for each regime.

V-DOMINATED CONVERGENCE

For $\alpha < 1$ convergence is dominated by $\mathbf{V}(t)$ and $\lambda_{\min}(\mathbf{\Lambda}(t)) \geq \frac{1}{\alpha^2} \lambda_{\min}(\mathbf{V}(t))$. We present the convergence theorem for the $\mathbf{V}$-dominated regime here.

**Theorem 4.1** (V-dominated convergence). *Suppose a neural network of the form (1.2) is initialized as in (2.4) with $\alpha \leq 1$ and that Assumptions 1,2 hold. In addition, suppose the neural network is trained via the regression loss (2.5) with targets $\mathbf{y}$ satisfying $\|\mathbf{y}\|_\infty = O(1)$. If $m = \Omega\big(n^4 \log(n/\delta)/\lambda_0^4\big)$, then with probability $1 - \delta$,*

1. *For iterations $s = 0, 1, \ldots, K$, the evolution matrix $\mathbf{\Lambda}(s)$ satisfies $\lambda_{\min}(\mathbf{\Lambda}(s)) \geq \frac{\lambda_0}{2\alpha^2}$.*

2. *WeightNorm training with gradient descent of step-size $\eta = O\left(\frac{\alpha^2}{\|\mathbf{V}^\infty\|_2}\right)$ converges linearly as*

$$\|\mathbf{f}(s) - \mathbf{y}\|_2^2 \leq \left(1 - \frac{\eta \lambda_0}{2\alpha^2}\right)^s \|\mathbf{f}(0) - \mathbf{y}\|_2^2.$$

The proof of Theorem 4.1 is presented in Appendix C. We will provide a sketch below. We make the following observations about our $\mathbf{V}$-dominated convergence result.

The required over-parameterization $m$ is independent of $\alpha$. Further, the dependence of $m$ on the failure probability is $\log(1/\delta)$. This improves previous results that require polynomial dependence of order $\delta^3$. Additionally, we reduce the dependence on the sample size from $n^6$ (as appears in [3]) to $n^4 \log(n)$.

In Theorem 4.1, smaller $\alpha$ leads to faster convergence, since the convergence is dictated by $\lambda_0/\alpha^2$. Nonetheless, smaller $\alpha$ is also at the cost of smaller allowed step-sizes, since $\eta = O(\alpha^2/\|\mathbf{V}^\infty\|_2)$. The trade-off between step-size and convergence speed is typical. For example, this is implied in Chizat et al. [12], where nonetheless the authors point out that for gradient flow training, the increased convergence rate is not balanced by a limitation on the step-size. The works [4, 19, 32] define an effective step-size (adaptive step-size) $\eta' = \eta/\alpha^2$ to avoid the dependence of $\eta$ on $\alpha$.

G-DOMINATED CONVERGENCE

For $\alpha > 1$ our convergence result for the class (1.2) is based on monitoring the least eigenvalue of $\mathbf{G}(t)$. Unlike $\mathbf{V}$-dominated convergence, $\alpha$ does not affect the convergence speed in this regime.

**Theorem 4.2** (G-dominated convergence). *Suppose a network of the form (1.2) is initialized as in (2.4) with $\alpha \geq 1$ and that Assumptions 1, 2 hold. In addition, suppose the neural network is trained via the regression loss (2.5) with targets $\mathbf{y}$ satisfying $\|\mathbf{y}\|_\infty = O(1)$. If $m = \Omega\big(\max\{n^4 \log(n/\delta)/\alpha^4 \mu_0^4, n^2 \log(n/\delta)/\mu_0^2\}\big)$, then with probability $1 - \delta$,*

1. *For iterations $s = 0, 1, \ldots, K$, the evolution matrix $\mathbf{\Lambda}(s)$ satisfies $\lambda_{\min}(\mathbf{\Lambda}(s)) \geq \frac{\mu_0}{2}$.*

2. *WeightNorm training with gradient descent of step-size $\eta = O\left(\frac{1}{\|\mathbf{\Lambda}(t)\|}\right)$ converges linearly as*

$$\|\mathbf{f}(s) - \mathbf{y}\|_2^2 \le \left(1 - \frac{\eta\mu_0}{2}\right)^s \|\mathbf{f}(0) - \mathbf{y}\|_2^2.$$

We make the following observations about our **G**-dominated convergence result, and provide a proof sketch further below.

Theorem 4.2 holds for $\alpha \ge 1$ so long as $m = \Omega\left(\max\left\{n^4 \log(n/\delta)/\mu_0^4\alpha^4, n^2 \log(n/\delta)/\mu_0^2\right\}\right)$. Taking $\alpha = \sqrt{n/\mu_0}$ gives an optimal required over-parameterization of order

$$m = \Omega\left(n^2 \log(n/\delta)/\mu_0^2\right).$$

This significantly improves on previous results [15] for un-normalized training that have dependencies of order 4 in the least eigenvalue, cubic dependence in $1/\delta$, and $n^6$ dependence in the number of samples $n$. In contrast to **V**-dominated convergence, here the rate of convergence $\mu_0$ is independent of $\alpha$ but the over-parameterization $m$ is $\alpha$-dependent. We elaborate on this curious behavior in the next sections.

**Proof sketch of main results**    The proof of Theorems 4.1 and 4.2 is inspired by a series of works including [3, 13, 15, 33, 36]. The proof has the following steps: (**I**) We show that at initialization $\mathbf{V}(0), \mathbf{G}(0)$ can be viewed as empirical estimates of averaged data-dependent kernels $\mathbf{V}^\infty, \mathbf{G}^\infty$ that are strictly positive definite under Assumption 1. (**II**) For each regime, we prove that the corresponding kernel remains positive definite if $\mathbf{v}_k(t)$ and $g_k(t)$ remain near initialization for each $1 \le k \le m$. (**III**) Given a uniformly positive definite evolution matrix $\mathbf{\Lambda}(t)$ and sufficient over-parameterization we show that each neuron, $\mathbf{v}_k(t), g_k(t)$ remains close to its initialization. The full proof is presented in Appendix B for gradient flow and Appendix C for finite-step gradient descent.

While the spirit of the proof is familiar from other works, the different convergence regimes explain the utility of WeightNorm. We elaborate further on this in the discussion.

## 5    RELATED WORK

**Dynamic normalization theory**    A number of recent works attempt to explain the dynamics and utility of dynamic normalization methods. The original works [20, 28] of BatchNorm and Weight-Norm resp. suggest that dynamic normalization methods improve training by fixing the intermediate layers' output distributions. The works of Bjorck et al. [8] and Santurkar et al. [29] argue that BatchNorm may improve optimization by improving smoothness of the Hessian of the loss, therefore allowing for larger step-sizes with reduced instability. Hoffer et al. [18] showed that the effective step-size in BatchNorm is divided by the magnitude of the weights, this followed the work of WNgrad [32] that introduces an adaptive step-size algorithm based on this fact. Following the intuition of WNGrad, Arora et al. [4] proved that for smooth loss and network functions, the diminishing "effective step-size" of normalization methods lead to convergence with optimal convergence rate for properly initialized step-sizes. The work of [22] explains the accelerated convergence of BatchNorm from a "length-direction decoupling" perspective. The authors along with [9] analyze the linear least squares regime, with [22] presenting a bisection method for finding the optimal weights. Robustness and regularization of Batch Normalization is investigated in [25] and improved generalization is analyzed empirically. Shortly after the original work of WeightNorm, [35] showed that for a single precptron WeightNorm may speed-up training and emphasized the importance of the norm of the initial weights. Additional stability properties were studied by [34] via mean-field analysis. The authors show that gradient instability is inevitable even with BatchNorm as the number of layers increases; this is in agreement with [7] for networks with residual connections. The work of [5] suggests initialization strategies for WeightNorm and derives lower bounds on the width to guarantee same order gradients across the layers.

**Over-parametrized neural networks**    There has been a significant recent literature studying the convergence of un-normalized over-parametrized neural networks. The analysis of the majority of the works relies on the width of the layers. These include 2-layer networks trained with Gaussian inputs and outputs from a teacher network [30], [24] and [14] (with WeightNorm). Assumptions on the data distribution are relaxed in [15] and the works that followed [3, 33, 36]. Our work is inspired by the mechanism presented in this chain of works. Wu et al. [33] extend convergence results to adaptive

step-size methods and propose AdaLoss. Recently, the global convergence of over-parameterized neural networks was also extended to deep architectures [2, 13, 37, 38]. In the over-parameterized regimes, Arora et al. [3] develop generalization properties for the networks of the form (1.1). In addition, in the context of generalization, Allen-Zhu et al. [1] illustrates good generalization for deep neural networks trained with gradient descent. Cao and Gu [10] and [11] derive generalization error bound of gradient descent and stochastic gradient descent for learning over-parameterization deep ReLU neural networks.

## 6 DISCUSSION

In this section we interpret our main results and make a connection of the convergence theory with "lazy training".

Chizat et al. [12] analyze recent works of over-parameterized convergence based on the NTK and note that re-scaling the network's outputs during gradient flow training leads to fast convergence while at the same time the weights remain close to their initialization. This is interpreted as a "linear solution", meaning that the direction of the weights does not change at training. The authors also refer to this type of convergence as "lazy training".

In our $\mathbf{G}$-dominated convergence, we observe a new type of "lazy training" that is different from the one presented in [12] . We refer to our $\mathbf{G}$ regime as "lazy training" since the directions of the weights remain fairly stationary. Nonetheless, in $\mathbf{G}$-dominated "lazy training", the magnitudes of the weights $(g_k)$ change. Moreover, we observe this phenomenon not only in the continuous flow setting but also in the finite step gradient descent setting. Unlike [12] where it is argued that neural networks do not necessarily follow the "lazy training" regime, we believe that $\mathbf{G}$-dominated convergence actually is common but that it emerges at later stages in training, after the weights have adopted their directions in the $\mathbf{V}$-dominated regime and improves stability.

Overall training with WeightNorm leads to a gradual transition from $\mathbf{V}$-dominated to the $\mathbf{G}$-dominated regimes. We recall that since the weights grow (see (2.3)), the WeightNorm coefficient $\alpha$ increases during training progressively. The direction of the weights changes rapidly at the earlier stages of training when $\alpha$ is small, and $\mathbf{G}$-dominated convergence ensues as $\alpha$ grows, leading to improved stability. For $\alpha > 1$ this allows us to ease the requirements made on the over-parameterization (less is sufficient) and step size (a bigger step size is possible, of order $\eta = O(1/\|\mathbf{G}(t)\|_2)$). From this perspective, the utility of WeightNorm is attributed to the allowed larger step-sizes and increased stability at the later stages of training ($\mathbf{G}$-dominated), all while maintaining fast convergence at the beginning of training ($\mathbf{V}$-dominated convergence).

Dynamic normalization is the most common optimization set-up of current deep learning models, yet understanding the convergence of such optimization methods is still an open problem. In this work we present a proof giving sufficient conditions for convergence of dynamically normalized 2-layer ReLU networks trained with gradient descent. To the best of our knowledge this is the first proof showcasing convergence of gradient descent training of neural networks with dynamic normalization and general data, where the objective function is non-smooth and non-convex. The notion of "length-direction decoupling" is clarified by the neural tangent kernel $\mathbf{\Lambda}(t)$ that naturally separates in our analysis into "length", $\mathbf{G}(t)$, and "direction", $\mathbf{V}(t)/\alpha^2$, components. For $\alpha = 1$ the decomposition initially matches un-normalized training. Yet we discover that in general, normalized training with gradient descent leads to 2 regimes dominated by different pieces of the neural tangent kernel. We note that training typically commences in the $\mathbf{V}$-dominated regime and transitions into the $\mathbf{G}$-dominated regime as training proceeds and the magnitude of the weights grow. Our improved analysis is able to reduce the amount of over-parameterization that was needed in previous convergence works in the un-normalized setting and in the $\mathbf{G}$-dominated regime, we prove convergence with a significantly lower amount of over-parameterization as compared with un-normalized training.

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

APPENDIX

We present the detailed proofs of the main results of the paper below. The appendix is organized as follows. We provide proofs to the simple propositions regarding the NTK presented in the paper in Appendix A, and prove the main results for $\mathbf{V}$-dominated and $\mathbf{G}$-dominated convergence in the settings of gradient flow and gradient descent in Appendices B,C. The proofs for gradient flow and gradient descent share the same main idea, yet the proof for gradient descent has a considerate number of additional technicalities. In Appendices D and E we prove the lemmas used in the analysis of Appendices B and C respectively.

## A  WEIGHTNORM DYNAMICS PROOFS

In this section we provide proofs for Proposition 1, which describes the relation between vanilla and WeightNorm NTKs and Observation 1 of the paper.

**Proof of Proposition 1:**
We would like to show that $\mathbf{V}(0) + \mathbf{G}(0) = \mathbf{H}(0)$. For each entry, consider

$$(\mathbf{V}(0) + \mathbf{G}(0))_{ij} = \frac{1}{m} \sum_{k=1}^{m} \left\langle \mathbf{x}_i^{\mathbf{v}_k(0)^{\perp}}, \ \mathbf{x}_j^{\mathbf{v}_k(0)^{\perp}} \right\rangle \mathbb{1}_{ik}(0) \mathbb{1}_{jk}(0) + \frac{1}{m} \sum_{k=1}^{m} \left\langle \mathbf{x}_i^{\mathbf{v}_k(0)}, \ \mathbf{x}_j^{\mathbf{v}_k(0)} \right\rangle \mathbb{1}_{ik}(0) \mathbb{1}_{jk}(0).$$

Note that

$$\left\langle \mathbf{x}_i, \ \mathbf{x}_j \right\rangle = \left\langle \mathbf{x}_i^{\mathbf{v}_k(0)} + \mathbf{x}_i^{\mathbf{v}_k(0)^{\perp}}, \ \mathbf{x}_j^{\mathbf{v}_k(0)} + \mathbf{x}_j^{\mathbf{v}_k(0)^{\perp}} \right\rangle = \left\langle \mathbf{x}_i^{\mathbf{v}_k(0)^{\perp}}, \ \mathbf{x}_j^{\mathbf{v}_k(0)^{\perp}} \right\rangle + \left\langle \mathbf{x}_i^{\mathbf{v}_k(0)}, \ \mathbf{x}_j^{\mathbf{v}_k(0)} \right\rangle.$$

This gives

$$(\mathbf{V}(0) + \mathbf{G}(0))_{ij} = \frac{1}{m} \sum_{k=1}^{m} \left\langle \mathbf{x}_i, \ \mathbf{x}_j \right\rangle \mathbb{1}_{ik}(0) \mathbb{1}_{jk}(0) = \mathbf{H}_{ij}(0)$$

which proves the claim. $\qquad\square$

**Proof of Observation 1:**
We show that the initialization of the network is independent of $\alpha$. Take $\alpha, \beta > 0$, and for each $k$, initialize $\mathbf{v}_k^{\alpha}, \mathbf{v}_k^{\beta}$ as

$$\mathbf{v}_k^{\alpha}(0) \sim N(0, \alpha^2 \mathbf{I}), \quad \mathbf{v}_k^{\beta}(0) \sim N(0, \beta^2 \mathbf{I}).$$

Then

$$\frac{\mathbf{v}_k^{\alpha}(0)}{\|\mathbf{v}_k^{\alpha}(0)\|_2} \sim \frac{\mathbf{v}_k^{\beta}(0)}{\|\mathbf{v}_k^{\beta}(0)\|_2} \sim \text{Unif}(\mathcal{S}^{d-1}) \quad \text{(in distribution)}.$$

Hence the distribution of each neuron $\sigma\left(\frac{\mathbf{v}_k(0)}{\|\mathbf{v}_k(0)\|_2}\right)$ at initialization is independent of $\alpha$. Next for $g_k(0)$, we note that

$$\|\mathbf{v}_k^{\alpha}(0)\|_2 \sim \frac{\alpha}{\beta} \|\mathbf{v}_k^{\beta}(0)\|_2.$$

Initializing $g_k^{\alpha}(0), g_k^{\beta}(0)$ as in (2.4),

$$g_k^{\alpha}(0) = \frac{\|\mathbf{v}_k(0)\|_2}{\alpha}, \quad g_k^{\beta}(0) = \frac{\|\mathbf{v}_k(0)\|_2}{\beta},$$

gives

$$g_k^{\alpha}(0), \quad g_k^{\beta}(0) \sim \chi_d, \quad \text{and} \quad \frac{g_k^{\alpha}(0) \mathbf{v}_k^{\alpha}(0)}{\|\mathbf{v}_k^{\alpha}(0)\|_2} \sim \frac{g_k^{\beta}(0) \mathbf{v}_k^{\beta}(0)}{\|\mathbf{v}_k^{\beta}(0)\|_2} \sim N(0, \mathbf{I}),$$

for all $\alpha, \beta$. This shows that the network initialization is independent of $\alpha$ and is equivalent to the initialization of the un-normalized setting. Similarly, inspecting the terms in the summands of $\mathbf{V}(0), \mathbf{G}(0)$ shows that they are also independent of $\alpha$. For

$$\mathbf{V}_{ij}(0) = \frac{1}{m} \sum_{k=1}^{m} \mathbb{1}_{ik}(0) \mathbb{1}_{jk}(0) \left( \frac{\alpha c_k \cdot g_k(0)}{\|\mathbf{v}_k(0)\|_2} \right)^2 \left\langle \mathbf{x}_i^{\mathbf{v}_k(0)^{\perp}}, \ \mathbf{x}_j^{\mathbf{v}_k(0)^{\perp}} \right\rangle$$

the terms $\mathbb{1}_{ik}(0)$, $\mathbf{x}_i^{\mathbf{v}_k(0)^\perp}$ are independent of scale, and the fraction in the summand is identically 1. $\mathbf{G}(0)$ defined as

$$\mathbf{G}_{ij}(0) = \frac{1}{m} \sum_{k=1}^{m} \mathbb{1}_{ik}(0) \mathbb{1}_{jk}(0) \langle \mathbf{x}_i^{\mathbf{v}_k(0)}, \mathbf{x}_j^{\mathbf{v}_k(0)} \rangle$$

is also invariant of scale since the projection onto a vector direction $\mathbf{v}_k(0)$ is independent of scale. $\quad\square$

## B  CONVERGENCE PROOF FOR GRADIENT FLOW

In this section we derive the convergence results for gradient flow.

The main results are analogous to Theorems 4.1, 4.2 but by considering gradient flow instead of gradient descent the proofs are simplified. In Appendix C we prove the main results from Section 4 (Theorem 4.1, 4.2) for finite step gradient descent.

We state our convergence results for gradient flow.

**Theorem B.1** (**V**-dominated convergence). *Suppose a network from the class* (1.2) *is initialized as in* (2.4) *with $\alpha < 1$ and that assumptions 1,2 hold. In addition, suppose the neural network is trained via the regression loss* (2.5) *with target $\mathbf{y}$ satisfying $\|\mathbf{y}\|_\infty = O(1)$. Then if $m = \Omega\big(n^4 \log(n/\delta)/\lambda_0^4\big)$, WeightNorm training with gradient flow converges at a linear rate, with probability $1 - \delta$, as*

$$\|\mathbf{f}(t) - \mathbf{y}\|_2^2 \le \exp(-\lambda_0 t/\alpha^2)\|\mathbf{f}(0) - \mathbf{y}\|_2^2.$$

This theorem is analogous to Theorem 4.1 but since here, the settings are of gradient flow there is no mention of the step-size. It is worth noting that smaller $\alpha$ leads to faster convergence and appears to not affect the other hypotheses of the flow theorem. This "un-interuptted" fast convergence behavior does not extend to finite-step gradient descent where the increased convergence rate is balanced by decreasing the allowed step-size.

The second main result for gradient flow is for **G**-dominated convergence.

**Theorem B.2** (**G**-dominated convergence). *Suppose a network from the class* (1.2) *is initialized as in* (2.4) *with $\alpha > 1$ and that assumptions 1, 2 hold. In addition, suppose the neural network is trained on the regression loss* (2.5) *with target $\mathbf{y}$ satisfying $\|\mathbf{y}\|_\infty = O(1)$. Then if $m = \Omega\big(\max\big\{n^4 \log(n/\delta)/\alpha^4\mu_0^4, n^2 \log(n/\delta)/\mu_0^2\big\}\big)$, WeightNorm training with gradient flow converges at a linear rate, with probability $1 - \delta$, as*

$$\|\mathbf{f}(t) - \mathbf{y}\|_2^2 \le \exp(-\mu_0 t)\|\mathbf{f}(0) - \mathbf{y}\|_2^2.$$

### B.1  PROOF SKETCH

To prove the results above we follow the steps introduced in the proof sketch of Section 4. The main idea of the proofs for **V** and **G** dominated convergence are analogous and a lot of the proofs are based of Du et al. [15]. We show that in each regime, we attain linear convergence by proving that the least eigenvalue of the evolution matrix $\boldsymbol{\Lambda}(t)$ is strictly positive. For the **V**-dominated regime we lower bound the least eigenvalue of $\boldsymbol{\Lambda}(t)$ as $\lambda_{\min}(\boldsymbol{\Lambda}(t)) \ge \lambda_{\min}(\mathbf{V}(t))/\alpha^2$ and in the **G**-dominated regime we lower bound the least eigenvalue as $\lambda_{\min}(\boldsymbol{\Lambda}(t)) \ge \lambda_{\min}(\mathbf{G}(t))$.

The main part of the proof is showing that $\lambda_{\min}(\mathbf{V}(t)), \lambda_{\min}(\mathbf{G}(t))$ stay uniformly positive. We use several lemmas to show this claim.

In each regime, we first show that at initialization the kernel under consideration, $\mathbf{V}(0)$ or $\mathbf{G}(0)$, has a positive least eigenvalue. This is shown via concentration to an an auxiliary kernel (Lemmas B.1, B.2), and showing that the auxiliary kernel is also strictly positive definite (Lemma 4.1).

**Lemma B.1.** *Let $\mathbf{V}(0)$ and $\mathbf{V}^\infty$ be defined as in* (3.3) *and* (4.2), *assume the network width $m$ satisfies $m = \Omega\big(\frac{n^2 \log(n/\delta)}{\lambda_0^2}\big)$. Then with probability $1 - \delta$,*

$$\|\mathbf{V}(0) - \mathbf{V}^\infty\|_2 \le \frac{\lambda_0}{4}.$$

**Lemma B.2.** *Let* $\mathbf{G}(0)$ *and* $\mathbf{G}^\infty$ *be defined as in* (3.4) *and* (4.3)*, assume* $m$ *satisfies* $m = \Omega\left(\frac{n^2 \log(n/\delta)}{\mu_0^2}\right)$*. Then with probability* $1 - \delta$*,*

$$\|\mathbf{G}(0) - \mathbf{G}^\infty\|_2 \leq \frac{\mu_0}{4}.$$

After showing that $\mathbf{V}(0), \mathbf{G}(0)$ have a positive least-eigenvalue we show that $\mathbf{V}(t), \mathbf{G}(t)$ maintain this positive least eigenvalue during training. This part of the proof depends on the over-parameterization of the networks. The main idea is showing that if the individual parameters $\mathbf{v}_k(t), g_k(t)$ do not change too much during training, then $\mathbf{V}(t), \mathbf{G}(t)$ remain close enough to $\mathbf{V}(0), \mathbf{G}(0)$ so that they are still uniformly strictly positive definite. We prove the results for $\mathbf{V}(t)$ and $\mathbf{G}(t)$ separately since each regime imposes different restrictions on the trajectory of the parameters.

For now, in Lemmas B.3, B.4, B.5, we make assumptions on the parameters of the network not changing "too much"; later we show that this holds and is the result of over-parameterization. Specifically, over-parameterization ensures that the parameters stay at a small maximum distance from their initialization.

**V-dominated convergence** To prove the least eigenvalue condition on $\mathbf{V}(t)$, we introduce the surrogate Gram matrix $\hat{\mathbf{V}}(t)$ defined entry-wise as

$$\hat{\mathbf{V}}_{ij}(t) = \frac{1}{m} \sum_{k=1}^{m} \left\langle \mathbf{x}_i^{\mathbf{v}_k(t)^\perp}, \mathbf{x}_j^{\mathbf{v}_k(t)^\perp} \right\rangle \mathbb{1}_{ik}(t) \mathbb{1}_{jk}(t). \tag{B.1}$$

This definition aligns with $\mathbf{V}(t)$ if we replace the scaling term $\left(\frac{\alpha c_k g_k(t)}{\|\mathbf{v}_k(t)\|_2}\right)^2$ in each term in the sum $\mathbf{V}_{ij}(t)$ by 1.

To monitor $\mathbf{V}(t) - \mathbf{V}(0)$ we consider $\hat{\mathbf{V}}(t) - \mathbf{V}(0)$ and $\mathbf{V}(t) - \hat{\mathbf{V}}(t)$ in Lemmas B.3 and B.4 respectively:

**Lemma B.3** (Rectifier sign-changes)**.** *Suppose* $\mathbf{v}_1(0), \ldots, \mathbf{v}_k(0)$ *are sampled i.i.d. as* (2.4)*. In addition assume we have* $m = \Omega\left(\frac{(m/\delta)^{1/d} n \log(n/\delta)}{\alpha \lambda_0}\right)$ *and* $\|\mathbf{v}_k(t) - \mathbf{v}_k(0)\|_2 \leq \frac{\alpha \lambda_0}{96n(m/\delta)^{1/d}} =: R_v$*. Then with probability* $1 - \delta$*,*

$$\|\hat{\mathbf{V}}(t) - \mathbf{V}(0)\|_2 \leq \frac{\lambda_0}{8}.$$

**Lemma B.4.** *Define*

$$R_g = \frac{\lambda_0}{48n(m/\delta)^{1/d}}, \quad R_v = \frac{\alpha \lambda_0}{96n(m/\delta)^{1/d}}. \tag{B.2}$$

*Suppose the conditions of Lemma B.3 hold, and that* $\|\mathbf{v}_k(t) - \mathbf{v}_k(0)\|_2 \leq R_v$*,* $\|g_k(t) - g_k(0)\|_2 \leq R_g$ *for all* $1 \leq k \leq m$*. Then with probability* $1 - \delta$*,*

$$\|\mathbf{V}(t) - \mathbf{V}(0)\|_2 \leq \frac{\lambda_0}{4}.$$

**G-dominated convergence** We ensure that $\mathbf{G}(t)$ stays uniformly positive definite if the following hold.

**Lemma B.5.** *Given* $\mathbf{v}_1(0), \ldots, \mathbf{v}_k(0)$ *generated i.i.d. as in* (2.4)*, suppose that for each* $k$*,* $\|\mathbf{v}_k(t) - \mathbf{v}_k(0)\|_2 \leq \frac{\sqrt{2\pi}\alpha\mu_0}{8n(m/\delta)^{1/d}} =: \tilde{R}_v$*, then with probability* $1 - \delta$*,*

$$\|\mathbf{G}(t) - \mathbf{G}(0)\|_2 \leq \frac{\mu_0}{4}.$$

After deriving sufficient conditions to maintain a positive least eigenvalue at training, we restate the discussion of linear convergence from Section 4 formally.

**Lemma B.6.** *Consider the linear evolution* $\frac{d\mathbf{f}}{dt} = -\left(\mathbf{G}(t) + \frac{\mathbf{V}(t)}{\alpha^2}\right)(\mathbf{f}(t) - \mathbf{y})$ *from (3.5). Suppose that* $\lambda_{\min}\left(\mathbf{G}(t) + \frac{\mathbf{V}(t)}{\alpha^2}\right) \geq \frac{\omega}{2}$ *for all times* $0 \leq t \leq T$. *Then*

$$\|\mathbf{f}(t) - \mathbf{y}\|_2^2 \leq \exp(-\omega t)\|\mathbf{f}(0) - \mathbf{y}\|_2^2$$

*for all times* $0 \leq t \leq T$.

Using the linear convergence result of Lemma B.6, we can now bound the trajectory of the parameters from their initialization.

**Lemma B.7.** *Suppose that for all* $0 \leq t \leq T$, $\lambda_{\min}\left(\mathbf{G}(t) + \frac{1}{\alpha^2}\mathbf{V}(t)\right) \geq \frac{\omega}{2}$ *and* $|g_k(t) - g_k(0)| \leq R_g \leq 1/(m/\delta)^{1/d}$. *Then with probability* $1 - \delta$ *over the initialization*

$$\|\mathbf{v}_k(t) - \mathbf{v}_k(0)\|_2 \leq \frac{4\sqrt{n}\|\mathbf{f}(0) - \mathbf{y}\|_2}{\alpha\omega\sqrt{m}} =: R_v' \tag{B.3}$$

*for each* $k$ *and all times* $0 \leq t \leq T$.

**Lemma B.8.** *Suppose that for all* $0 \leq t \leq T$, $\lambda_{\min}\left(\mathbf{G}(t) + \frac{1}{\alpha^2}\mathbf{V}(t)\right) \geq \frac{\omega}{2}$. *Then with probability* $1 - \delta$ *over the initialization*

$$|g_k(t) - g_k(0)| \leq \frac{4\sqrt{n}\|\mathbf{f}(0) - \mathbf{y}\|_2}{\sqrt{m}\omega} =: R_g'$$

*for each* $k$ *and all times* $0 \leq t \leq T$.

The distance of the parameters from initialization depends on the convergence rate (which depends on $\lambda_{\min}(\boldsymbol{\Lambda}(t))$) and the width of the network $m$. We therefore are able to find sufficiently large $m$ for which the maximum parameter trajectories are not too large so that we have that the least eigenvalue of $\boldsymbol{\Lambda}(t)$ is bounded from 0; this proves the main claim.

Before proving the main results in the case of gradient flow, we use two more technical lemmas.

**Lemma B.9.** *Suppose that the network is initialized as (2.4) and that* $\mathbf{y} \in \mathbb{R}^n$ *has bounded entries* $|y_i| \leq M$. *Then* $\|\mathbf{f}(0) - \mathbf{y}\|_2 \leq C\sqrt{n\log(n/\delta)}$ *for some absolute constant* $C > 0$.

**Lemma B.10** (Failure over initialization). *Suppose* $\mathbf{v}_1(0), \ldots, \mathbf{v}_k(0)$ *are initialized i.i.d. as in (2.4) with input dimension* $d$. *Then with probability* $1 - \delta$,

$$\max_{k \in [m]} \frac{1}{\|\mathbf{v}_k(0)\|_2} \leq \frac{(m/\delta)^{1/d}}{\alpha}.$$

*In addition by (2.3), for all* $t \geq 0$, *with probability* $1 - \delta$,

$$\max_{k \in [m]} \frac{1}{\|\mathbf{v}_k(t)\|_2} \leq \frac{(m/\delta)^{1/d}}{\alpha}.$$

**Remark** (Assumption 2). *Predominately, machine learning applications reside in the high dimensional regime with* $d \geq 50$. *Typically* $d \gg 50$. *This therefore leads to an expression* $(m/\delta)^{1/d}$ *that is essentially constant. For example, if* $d = 50$, *for* $\max_{k \in [m]} \frac{1}{\|\mathbf{v}_k(0)\|_2} \geq 10$, *one would need* $m/\delta \geq 10^{80}$ *(the tail of* $\chi_d^2$ *also has a factor of* $(d/2)! \cdot 2^{d/2}$ *which makes the assumption even milder). The term* $(m/\delta)^{1/d}$ *therefore may be taken as a constant for practicality,*

$$\max_{k \in [m]} \frac{1}{\|\mathbf{v}_k(0)\|_2} \leq \frac{C}{\alpha}.$$

While we make Assumption 2 when presenting our final bounds, for transparency we do not use Assumption 2 during our analysis and apply it only when we present the final over-parameterization results to avoid the overly messy bound. Without the assumption the theory still holds yet the over-parameterization bound worsens by a power $1 + 1/(d-1)$. This is since the existing bounds can be modified, replacing $m$ with $m^{1-\frac{1}{d}}$.

**Proof of Theorem B.1:**
Note that $m = \Omega\big(n^4 \log(n/\delta)/\lambda_0^4\big)$ implies that Lemma B.1 holds. Further since the gradient flow updates are continous (since the time derivative with respect to each parameter may be bounded), there exist a small time $T$ for which the parameters are bounded from initialization for $0 \le t \le T$ and for all $1 \le k \le m$,

$$\|\mathbf{v}_k(t) - \mathbf{v}_k(0)\|_2 \le R_v, \ |g_k(t) - g_k(0)| \le R_g.$$

The bounded trajectory above, along with the over-parameterization, ensures that with probability $1 - \delta$ over the initialization that Lemma B.4 holds and that

$$\lambda_{\min}\left(\mathbf{G}(t) + \frac{1}{\alpha^2}\mathbf{V}(t)\right) \ge \lambda_{\min}(\mathbf{V}(t))/\alpha^2 \ge \frac{\lambda_0}{2\alpha^2}$$

for times $0 \le t \le T$. The condition on the eigenvalue of the evolution matrix along with the bounded trajectory of $g_k$ implies that Lemmas B.7, B.8 hold for at least times $0 \le t < T$. Define $T_0$ to be the first failure point of either Lemmas B.8, B.7, we have that $T_0 \ge T > 0$. For $0 \le t < T_0$, by substituting $m = \Omega\big(n^4 \log(n/\delta)/\lambda_0^4\big)$ and using the bound on $\|\mathbf{f}(0) - \mathbf{y}\|_2$ of Lemma B.9, a direct calculation utilizing Lemma B.7 shows that

$$\|\mathbf{v}_k(t) - \mathbf{v}_k(0)\|_2 \overset{\text{B.7}}{\le} \frac{\alpha\sqrt{n}\|\mathbf{f}(0) - \mathbf{y}\|_2}{\sqrt{m}\lambda_0} \le R_v.$$

Similarly by Lemma B.8 $m$ ensures that

$$|g_k(t) - g_k(0)| \overset{\text{B.8}}{\le} \frac{\alpha^2\sqrt{n}\|\mathbf{f}(0) - \mathbf{y}\|_2}{\sqrt{m}\lambda_0} \le R_g.$$

The over-parameterization of $m$ implies that the parameter trajectories stay close enough to initialization to satisfy the hypotheses of Lemmas B.3, B.4 and that $\lambda_{\min}(\mathbf{\Lambda}(t)) \ge \lambda_{\min}(\mathbf{V}(t))/\alpha^2 \ge \frac{\lambda_0}{2\alpha^2}$ at time $T_0$. This implies that Lemmas B.7, B.8 both hold at time $T_0$ which contradicts the definition of $T_0$. Therefore we conclude that Lemmas B.7, B.8 hold for $t > 0$ which implies that Lemma B.4 holds for $t > 0$ hence Lemma B.6 guarantees linear convergence. □

Here we consider the case where the convergence is dominated by $\mathbf{G}$. This occurs when $\alpha > 1$.
**Proof of Theorem B.2:**
Similarly to Theorem B.1, we note that $m = \Omega\big( \max \big\{ n^4 \log(n/\delta)/\alpha^4\mu_0^4, n^2 \log(n/\delta)/\mu_0^2 \big\}\big)$ implies that Lemma B.2 holds. Further since the gradient flow updates are continous (since the time derivative with respect to each parameter may be bounded), there exist small time $T$ for which the parameters are bounded from initialization for $0 \le t \le T$ and for all $1 \le k \le m$

$$\|\mathbf{v}_k(t) - \mathbf{v}_k(0)\|_2 \le \tilde{R}_v.$$

The bounded trajectory above, along with the over-parameterization, ensures that with probability $1 - \delta$ over the initialization that Lemma B.5 holds and that

$$\lambda_{\min}\left(\mathbf{G}(t) + \frac{1}{\alpha^2}\mathbf{V}(t)\right) \ge \lambda_{\min}(\mathbf{G}(t))/\alpha^2 \ge \frac{\mu_0}{2}$$

for times $0 \le t \le T$. The condition on the eigenvalue of the evolution matrix along with the bounded trajectory of $g_k$ implies that Lemmas B.7, B.8 hold for at least times $0 \le t < T$. Define $T_0$ to be the first failure point of either Lemmas B.8, B.7, we have that $T_0 \ge T > 0$. For $0 \le t < T_0$, by substituting $m = \Omega\big( \max \big\{ n^4 \log(n/\delta)/\alpha^4\mu_0^4, n^2 \log(n/\delta)/\mu_0^2 \big\}\big)$ and using the bound on $\|\mathbf{f}(0) - \mathbf{y}\|_2$ of Lemma B.9, a direct calculation utilizing Lemma B.7 shows that

$$\|\mathbf{v}_k(t) - \mathbf{v}_k(0)\|_2 \overset{\text{B.7}}{\le} \frac{4\sqrt{n}\|\mathbf{f}(0) - \mathbf{y}\|_2}{\alpha\mu_0\sqrt{m}} \overset{\text{B.9}}{\le} \frac{Cn\sqrt{\log(n/\delta)}}{\alpha\mu_0\sqrt{m}} \le \tilde{R}_v.$$

Similarly by Lemma B.8 $m$ ensures that

$$|g_k(t) - g_k(0)| \overset{\text{B.8}}{\le} \frac{\sqrt{n}\|\mathbf{f}(0) - \mathbf{y}\|_2}{\sqrt{m}\mu_0} \le R_g.$$

The over-parameterization of $m$ implies that the parameter trajectories stay close enough to initialization to satisfy the hypotheses of Lemmas B.5 and that $\lambda_{\min}(\mathbf{\Lambda}(t)) \geq \lambda_{\min}(\mathbf{G}(t))/\alpha^2 \geq \mu_0/2$ at time $T_0$. This implies that Lemmas B.7, B.8 both hold at time $T_0$ which contradicts the definition of $T_0$. Therefore we conclude that Lemmas B.7, B.8 hold for $t > 0$ which implies that Lemma B.5 holds for $t > 0$ hence Lemma B.6 guarantees linear convergence with rate $\mu_0/2$.

$\square$

Note that if $\alpha$ is large, the required complexity on $m$ is reduced. Taking $\alpha = \Omega(\sqrt{n/\mu_0})$ gives the improved bound

$$m = \Omega\left(\frac{n^2 \log(n/\delta)}{\mu_0^2}\right).$$

## C  FINITE STEP-SIZE TRAINING

The general technique of proof for gradient flow extends to finite-step gradient descent. Nonetheless, proving convergence for WeightNorm gradient descent exhibits additional complexities arising from the discrete updates and joint training with the new parameterization (1.2). We first introduce some needed notation.

Define $S_i(R)$ as the set of indices $k \in [m]$ corresponding to neurons that are close to the activity boundary of ReLU at initialization for a data point $\mathbf{x}_i$,

$$S_i(R) := \{k \in [m] : \exists \mathbf{v} \text{ with } \|\mathbf{v} - \mathbf{v}_k(0)\|_2 \leq R \text{ and } \mathbb{1}_{ik}(0) \neq \mathbb{1}\{\mathbf{v}^\top \mathbf{x}_i \geq 0\}\}.$$

We upper bound the cardinality of $|S_i(R)|$ with high probability.

**Lemma C.1.** *With probability $1 - \delta$, we have that for all $i$*

$$|S_i(R)| \leq \frac{\sqrt{2}mR}{\sqrt{\pi}\alpha} + \frac{16\log(n/\delta)}{3}.$$

Next we review some additional lemmas needed for the proof of Theorems 4.1, 4.2. Analogous to Lemmas B.7, B.8, we bound the finite-step parameter trajectories in Lemmas C.2, C.3.

**Lemma C.2.** *Suppose the norm of $\|\mathbf{f}(s) - \mathbf{y}\|_2^2$ decreases linearly for some convergence rate $\omega$ during gradient descent training for all iteration steps $s = 0, 1, \ldots, K$ with step-size $\eta$ as $\|\mathbf{f}(s) - \mathbf{y}\|_2^2 \leq (1 - \frac{\eta\omega}{2})^s \|\mathbf{f}(0) - \mathbf{y}\|_2^2$. Then for each $k$ we have*

$$|g_k(s) - g_k(0)| \leq \frac{4\sqrt{n}\|\mathbf{f}(0) - \mathbf{y}\|_2}{\sqrt{m}\omega}$$

*for iterations $s = 0, 1, \ldots, K + 1$.*

**Lemma C.3.** *Under the assumptions of Lemma C.2, suppose in addition that $|g_k(s) - g_k(0)| \leq 1/(m/\delta)^{1/d}$ for all iterations steps $s = 0, 1, \ldots K$. Then for each $k$,*

$$\|\mathbf{v}_k(s) - \mathbf{v}_k(0)\|_2 \leq \frac{8\sqrt{n}\|\mathbf{f}(0) - \mathbf{y}\|_2}{\alpha\sqrt{m}\omega}$$

*for $s = 0, 1, \ldots, K + 1$.*

To prove linear rate of convergence we analyze the $s + 1$ iterate error $\|\mathbf{f}(s + 1) - \mathbf{y}\|_2$ relative to that of the $s$ iterate, $\|\mathbf{f}(s) - \mathbf{y}\|_2$. Consider the network's coordinate-wise difference in output between iterations, $f_i(s + 1) - f_i(s)$, writing this explicitly based on gradient descent updates yields

$$f_i(s+1) - f_i(s) = \frac{1}{\sqrt{m}} \sum_{k=1}^{m} \frac{c_k g_k(s+1)}{\|\mathbf{v}_k(s+1)\|_2} \sigma(\mathbf{v}_k(s+1)^\top \mathbf{x}_i) - \frac{c_k g_k(s)}{\|\mathbf{v}_k(s)\|_2} \sigma(\mathbf{v}_k(s)^\top \mathbf{x}_i). \quad \text{(C.1)}$$

We now decompose the summand in (C.1) looking at the updates in each layer, $f_i(s+1) - f_i(s) = a_i(s) + b_i(s)$ with

$$a_i(s) = \frac{1}{\sqrt{m}} \sum_{k=1}^{m} \frac{c_k g_k(s+1)}{\|\mathbf{v}_k(s+1)\|_2} \sigma(\mathbf{v}_k(s)^\top \mathbf{x}_i) - \frac{c_k g_k(s)}{\|\mathbf{v}_k(s)\|_2} \sigma(\mathbf{v}_k(s)^\top \mathbf{x}_i),$$

$$b_i(s) = \frac{1}{\sqrt{m}} \sum_{k=1}^{m} \frac{c_k g_k(s+1)}{\|\mathbf{v}_k(s+1)\|_2} \big( \sigma(\mathbf{v}_k(s+1)^\top \mathbf{x}_i) - \sigma(\mathbf{v}_k(s)^\top \mathbf{x}_i) \big).$$

Further, each layer summand is then subdivided into a primary term and a residual. $a_i(s)$, corresponding to the difference in the first layer $\left( \frac{c_k g_k(s+1)}{\|\mathbf{v}_k(s+1)\|_2} - \frac{c_k g_k(s)}{\|\mathbf{v}_k(s)\|_2} \right)$, is subdivided into $a_i^I(s)$ and $a_i^{II}(s)$ as follows:

$$a_i^I(s) = \frac{1}{\sqrt{m}} \sum_{k=1}^{m} \left( \frac{c_k g_k(s+1)}{\|\mathbf{v}_k(s)\|_2} - \frac{c_k g_k(s)}{\|\mathbf{v}_k(s)\|_2} \right) \sigma(\mathbf{v}_k(s)^\top \mathbf{x}_i), \tag{C.2}$$

$$a_i^{II}(s) = \frac{1}{\sqrt{m}} \sum_{k=1}^{m} \left( \frac{c_k g_k(s+1)}{\|\mathbf{v}_k(s+1)\|_2} - \frac{c_k g_k(s+1)}{\|\mathbf{v}_k(s)\|_2} \right) \sigma(\mathbf{v}_k(s)^\top \mathbf{x}_i). \tag{C.3}$$

$b_i(s)$ is sub-divided based on the indices in the set $S_i$ that monitor the changes of the rectifiers. For now, $S_i = S_i(R)$ with $R$ to be set later in the proof. $b_i(s)$ is partitioned to summands in the set $S_i$ and the complement set,

$$b_i^I(s) = \frac{1}{\sqrt{m}} \sum_{k \notin S_i} \frac{c_k g_k(s+1)}{\|\mathbf{v}_k(s+1)\|_2} \big( \sigma(\mathbf{v}_k(s+1)^\top \mathbf{x}_i) - \sigma(\mathbf{v}_k(s)^\top \mathbf{x}_i) \big),$$

$$b_i^{II}(s) = \frac{1}{\sqrt{m}} \sum_{k \in S_i} \frac{c_k g_k(s+1)}{\|\mathbf{v}_k(s+1)\|_2} \big( \sigma(\mathbf{v}_k(s+1)^\top \mathbf{x}_i) - \sigma(\mathbf{v}_k(s)^\top \mathbf{x}_i) \big).$$

With this sub-division in mind, the terms corresponding to convergence are $\mathbf{a}^I(s), \mathbf{b}^I(s)$ whereas $\mathbf{a}^{II}(s), \mathbf{b}^{II}(s)$ are residuals that are the result of discretization. We define the primary and residual vectors $\mathbf{p}(s), \mathbf{r}(s)$ as

$$\mathbf{p}(s) = \frac{\mathbf{a}_I(s) + \mathbf{b}_I(s)}{\eta}, \quad \mathbf{r}(s) = \frac{\mathbf{a}_{II} + \mathbf{b}_{II}(s)}{\eta}. \tag{C.4}$$

If the residual $\mathbf{r}(s)$ is sufficiently small and $\mathbf{p}(s)$ may be written as $\mathbf{p}(s) = -\mathbf{\Lambda}(s)(\mathbf{f}(s) - \mathbf{y})$ for some iteration dependent evolution matrix $\mathbf{\Lambda}(s)$ that has

$$\lambda_{\min}(\mathbf{\Lambda}(s)) = \omega/2 \tag{C.5}$$

for $\omega > 0$ then the neural network (1.2) converges linearly when trained with WeightNorm gradient descent of step size $\eta$. We formalize the condition on $\mathbf{r}(s)$ below and later derive the conditions on the over-parameterization ($m$) ensuring that $\mathbf{r}(s)$ is sufficiently small.

**Property 1.** *Given a network from the class (1.2) initialized as in (2.4) and trained with gradient descent of step-size $\eta$, define the residual $\mathbf{r}(s)$ as in (C.4) and take $\omega$ as in (C.5). We specify the "residual condition" at iteration $s$ as*

$$\|\mathbf{r}(s)\|_2 \leq c\omega \|\mathbf{f}(s) - \mathbf{y}\|_2$$

*for a sufficiently small constant $c > 0$ independent of the data or initialization.*

Here we present Theorem C.1 which is the backbone of Theorems 4.1 and 4.2.

**Theorem C.1.** *Suppose a network from the class (1.2) is trained via WeightNorm gradient descent with an evolution matrix $\mathbf{\Lambda}(s)$ as in (C.5) satisfying $\lambda_{\min}(\mathbf{\Lambda}(s)) \geq \omega/2$ for $s = 0, 1, \ldots K$. In addition if the data meets assumptions 1, 2, the step-size $\eta$ of gradient descent satisfies $\eta \leq \frac{1}{3\|\mathbf{\Lambda}(s)\|_2}$ and that the residual $\mathbf{r}(s)$ defined in (C.4) satisfies Property 1 for $s = 0, 1, \ldots, K$ then we have that*

$$\|\mathbf{f}(s) - \mathbf{y}\|_2^2 \leq \left( 1 - \frac{\eta\omega}{2} \right)^s \|\mathbf{f}(0) - \mathbf{y}\|_2^2$$

*for $s = 0, 1, \ldots, K$.*

**Proof of Theorem C.1:**
This proof provides the foundation for the main theorems. In the proof we also derive key bounds to be used in Theorems 4.1, 4.2. We use the decomposition we described above and consider again the difference between consecutive terms $\mathbf{f}(s+1) - \mathbf{f}(s)$,

$$f_i(s+1) - f_i(s) = \frac{1}{\sqrt{m}} \sum_{k=1}^{m} \frac{c_k g_k(s+1)}{\|\mathbf{v}_k(s+1)\|_2} \sigma(\mathbf{v}_k(s+1)^\top \mathbf{x}_i) - \frac{c_k g_k(s)}{\|\mathbf{v}_k(s)\|_2} \sigma(\mathbf{v}_k(s)^\top \mathbf{x}_i). \quad \text{(C.6)}$$

Following the decompostion introduced in (C.2), $a_i^I(s)$ is re-written in terms of $\mathbf{G}(s)$,

$$
\begin{aligned}
a_i^I(s) &= \frac{1}{\sqrt{m}} \sum_{k=1}^{m} \frac{c_k}{\|\mathbf{v}_k(s)\|_2} \left( -\eta \frac{\partial L(s)}{\partial g_k} \right) \sigma(\mathbf{v}_k(s)^\top \mathbf{x}_i) \\
&= -\frac{\eta}{m} \sum_{k=1}^{m} \frac{c_k}{\|\mathbf{v}_k(s)\|_2} \sum_{j=1}^{n} (f_j(s) - y_j) \frac{c_k}{\|\mathbf{v}_k(s)\|_2} \sigma(\mathbf{v}_k^\top(s)\mathbf{x}_j) \sigma(\mathbf{v}_k^\top(s)\mathbf{x}_i) \\
&= -\eta \sum_{j=1}^{n} (f_j(s) - y_j) \frac{1}{m} \sum_{k=1}^{m} (c_k)^2 \sigma\left( \frac{\mathbf{v}_k(s)^\top \mathbf{x}_i}{\|\mathbf{v}_k(s)\|_2} \right) \sigma\left( \frac{\mathbf{v}_k(s)^\top \mathbf{x}_j}{\|\mathbf{v}_k(s)\|_2} \right) \\
&= -\eta \sum_{j=1}^{n} (f_j(s) - y_j) \mathbf{G}_{ij}(s),
\end{aligned}
$$

where the first equality holds by the gradient update rule $g_k(s+1) = g_k(s) - \eta \nabla_{g_k} L(s)$. In this proof we also derive bounds on the residual terms of the decomposition which we will aid us in the proofs of Theorems 4.1, 4.2. $a_i^I(s)$ is the primary term of $a_i(s)$, now we bound the residual term $a_i^{II}(s)$. Recall $a_i^{II}(s)$ is written as

$$a_i^{II}(s) = \frac{1}{\sqrt{m}} \sum_{k=1}^{m} \left( \frac{c_k g_k(s+1)}{\|\mathbf{v}_k(s+1)\|_2} - \frac{c_k g_k(s+1)}{\|\mathbf{v}_k(s)\|_2} \right) \sigma(\mathbf{v}_k(s)^\top \mathbf{x}_i),$$

which corresponds to the difference in the normalization in the second layer. Since $\nabla_{\mathbf{v}_k} L(s)$ is orthogonal to $\mathbf{v}_k(s)$ we have that

$$
\begin{aligned}
&c_k g_k(s+1) \left( \frac{1}{\|\mathbf{v}_k(s+1)\|_2} - \frac{1}{\|\mathbf{v}_k(s)\|_2} \right) \sigma(\mathbf{v}_k(s)^\top \mathbf{x}_i) \\
&= c_k g_k(s+1) \left( \frac{1}{\sqrt{\|\mathbf{v}_k(s)\|_2^2 + \eta^2 \|\nabla_{\mathbf{v}_k} L(s)\|_2^2}} - \frac{1}{\|\mathbf{v}_k(s)\|_2} \right) \sigma(\mathbf{v}_k(s)^\top \mathbf{x}_i) \\
&= \frac{-c_k g_k(s+1) \eta^2 \|\nabla_{\mathbf{v}_k} L(s)\|_2^2}{\|\mathbf{v}_k(s+1)\|_2 \|\mathbf{v}_k(s)\|_2 (\|\mathbf{v}_k(s)\|_2 + \|\mathbf{v}_k(s+1)\|_2)} \sigma(\mathbf{v}_k(s)^\top \mathbf{x}_i) \\
&\leq \frac{-c_k g_k(s+1) \eta^2 \|\nabla_{\mathbf{v}_k} L(s)\|_2^2}{2\|\mathbf{v}_k(s)\|_2 \|\mathbf{v}_k(s+1)\|_2} \sigma\left( \frac{\mathbf{v}_k(s)^\top \mathbf{x}_i}{\|\mathbf{v}_k(s)\|_2} \right),
\end{aligned}
$$

where the first equality above is by completing the square, and the inequality is due to the increasing magnitudes of $\|\mathbf{v}_k(s)\|_2$.

Since $0 \leq \sigma\left( \frac{\mathbf{v}_k(s)^\top \mathbf{x}_i}{\|\mathbf{v}_k(s)\|_2} \right) \leq 1$, the above can be bounded as

$$
\begin{aligned}
|a_i^{II}(s)| &\leq \frac{1}{\sqrt{m}} \sum_{k=1}^{m} \left| \frac{g_k(s+1) \eta^2 \|\nabla_{\mathbf{v}_k} L(s)\|_2^2}{2\|\mathbf{v}_k(s)\|_2 \|\mathbf{v}_k(s+1)\|_2} \right| \\
&\leq \frac{1}{\sqrt{m}} \sum_{k=1}^{m} \frac{\eta^2 \left(1 + R_g (m/\delta)^{1/d}\right)^3 n \|\mathbf{f}(s) - \mathbf{y}\|_2^2 (m/\delta)^{1/d}}{\alpha^4 m} \\
&= \frac{\eta^2 n \left(1 + R_g (m/\delta)^{1/d}\right)^3 \|\mathbf{f}(s) - \mathbf{y}\|_2^2 (m/\delta)^{1/d}}{\alpha^4 \sqrt{m}}. \quad \text{(C.7)}
\end{aligned}
$$

The second inequality is the result of applying the bound in equation (E.1) on the gradient norm $\|\nabla_{\mathbf{v}_k} L(s)\|_2$ and using Lemma B.10.

Next we analyze $b_i(s)$ and sub-divide it based on the sign changes of the rectifiers. Define the set $S_i := S_i(R)$ as in Lemma C.1 with $R$ taken to be such that $\|\mathbf{v}_k(s+1) - \mathbf{v}_k(0)\|_2 \leq R$ for all $k$. Take $b_i^{II}(s)$ as the sub-sum of $b_i(s)$ with indices $k$ from the set $S_i$.

$b_i^I(s)$ corresponds to the sub-sum with the remaining indices. By the definition of $S_i$, for $k \notin S_i$ we have that $\mathbb{1}_{ik}(s+1) = \mathbb{1}_{ik}(s)$. This enables us to factor $\mathbb{1}_{ik}(s)$ and represent $b_i^I(s)$ as a Gram matrix similar to $\mathbf{V}(s)$ with a correction term from the missing indices in $S_i$.

$$b_i^I(s) = -\frac{1}{\sqrt{m}} \sum_{k \notin S_i} \left( \frac{c_k g_k(s+1)}{\|\mathbf{v}_k(s+1)\|_2} \right) (\eta \langle \nabla_{\mathbf{v}_k} L(s), \mathbf{x}_i \rangle) \mathbb{1}_{ik}(s)$$

$$= -\frac{\eta}{m} \sum_{k \notin S_i} \left( \frac{c_k g_k(s+1)}{\|\mathbf{v}_k(s+1)\|_2} \right) \left( \frac{c_k g_k(s)}{\|\mathbf{v}_k(s)\|_2} \right) \sum_{j=1}^n (f_j(s) - y_j) \mathbb{1}_{ik}(s) \mathbb{1}_{jk}(s) \langle \mathbf{x}_j^{\mathbf{v}_k(s)^\perp}, \mathbf{x}_i \rangle.$$

Note that $\langle \mathbf{x}_j^{\mathbf{v}_k(s)^\perp}, \mathbf{x}_i \rangle = \langle \mathbf{x}_j^{\mathbf{v}_k(s)^\perp}, \mathbf{x}_i^{\mathbf{v}_k(s)^\perp} \rangle$ therefore,

$$b_i^I(s) = -\frac{\eta}{m} \sum_{k \notin S_i} \left( \frac{c_k g_k(s+1)}{\|\mathbf{v}_k(s+1)\|_2} \right) \left( \frac{c_k g_k(s)}{\|\mathbf{v}_k(s)\|_2} \right) \sum_{j=1}^n (f_j(s) - y_j) \mathbb{1}_{ik}(s) \mathbb{1}_{jk}(s) \langle \mathbf{x}_j^{\mathbf{v}_k(s)^\perp}, \mathbf{x}_i^{\mathbf{v}_k(s)^\perp} \rangle.$$

Define $\tilde{\mathbf{V}}(s)$ as

$$\tilde{\mathbf{V}}_{ij}(s) = \frac{1}{m} \sum_{k=1}^m \left( \frac{\alpha c_k g_k(s+1)}{\|\mathbf{v}_k(s+1)\|_2} \right) \left( \frac{\alpha c_k g_k(s)}{\|\mathbf{v}_k(s)\|_2} \right) \mathbb{1}_{jk}(s) \mathbb{1}_{ik}(s) \langle \mathbf{x}_i^{\mathbf{v}_k(s)^\perp}, \mathbf{x}_j^{\mathbf{v}_k(s)^\perp} \rangle.$$

This matrix is identical to $\mathbf{V}(s)$ except for a modified scaling term $\left( \frac{c_k^2 g_k(s+1) g_k(s)}{\|\mathbf{v}_k(s)\|_2 \|\mathbf{v}_k(s+1)\|_2} \right)$. We note however that

$$\min \left( \left( \frac{c_k g_k(s)}{\|\mathbf{v}_k(s)\|_2} \right)^2, \left( \frac{c_k g_k(s+1)}{\|\mathbf{v}_k(s+1)\|_2} \right)^2 \right) \leq \left( \frac{c_k g_k(s)}{\|\mathbf{v}_k(s)\|_2} \right) \left( \frac{c_k g_k(s+1)}{\|\mathbf{v}_k(s+1)\|_2} \right)$$

$$\leq \max \left( \left( \frac{c_k g_k(s)}{\|\mathbf{v}_k(s)\|_2} \right)^2, \left( \frac{c_k g_k(s+1)}{\|\mathbf{v}_k(s+1)\|_2} \right)^2 \right)$$

because $g_k(s), c_k^2$ are positive. Hence the matrix $\tilde{\mathbf{V}}(s)$ satisfies the hypothesis of Lemma B.4 entirely. We write $b_i^I(s)$ as

$$b_i^I(s) = -\eta/\alpha^2 \sum_{j=1}^n (f_j(s) - y_j)(\tilde{\mathbf{V}}_{ij}(s) - \tilde{\mathbf{V}}_{ij}^\perp(s)),$$

where we have defined

$$\tilde{\mathbf{V}}_{ij}^\perp(s) = \frac{1}{m} \sum_{k \in S_i} \left( \frac{\alpha c_k g_k(s)}{\|\mathbf{v}_k(s)\|_2} \right) \left( \frac{\alpha c_k g_k(s+1)}{\|\mathbf{v}_k(s+1)\|_2} \right) \mathbb{1}_{ik}(s) \mathbb{1}_{jk}(s) \langle \mathbf{x}_i^{\mathbf{v}_k(s)^\perp}, \mathbf{x}_j^{\mathbf{v}_k(s)^\perp} \rangle. \tag{C.8}$$

We then bound the magnitude of each entry $\tilde{\mathbf{V}}_{ij}^\perp(s)$:

$$\tilde{\mathbf{V}}_{ij}^\perp(s) = \frac{1}{m} \sum_{k \in S_i} \left( \frac{\alpha c_k g_k(s)}{\|\mathbf{v}_k(s)\|_2} \right) \left( \frac{\alpha c_k g_k(s+1)}{\|\mathbf{v}_k(s+1)\|_2} \right) \mathbb{1}_{ik}(s) \mathbb{1}_{jk}(s) \langle \mathbf{x}_i^{\mathbf{v}_k(s)^\perp}, \mathbf{x}_j^{\mathbf{v}_k(s)^\perp} \rangle$$

$$\leq \frac{(1 + R_g(m/\delta)^{1/d})^2 |S_i|}{m}. \tag{C.9}$$

Lastly we bound the size of the residual term $b_i^{II}(s)$,

$$
\begin{aligned}
|b_i^{II}(s)| &= \left| -\frac{1}{\sqrt{m}} \sum_{k \in S_i} \frac{c_k g_k(s+1)}{\|\mathbf{v}_k(s+1)\|_2} \left( \sigma(\mathbf{v}_k(s+1)^\top \mathbf{x}_i) - \sigma(\mathbf{v}_k(s)^\top \mathbf{x}_i) \right) \right| \\
&\leq \frac{g_k(s+1)\eta|S_i| \cdot \|\nabla_{\mathbf{v}_k} L(s)\|_2}{\sqrt{m}\|\mathbf{v}_k(s+1)\|_2} \\
&\leq \frac{\eta|S_i|(1+(m/\delta)^{1/d}R_g)\|\nabla_{\mathbf{v}_k} L(s)\|_2}{\alpha\sqrt{m}}.
\end{aligned}
$$

Where we used the Lipschitz continuity of $\sigma$ in the first bound, and took $R_g > 0$ that satisfies $|g_k(s+1) - g_k(0)| \leq R_g$ in the second inequality. Applying the bound (E.1),

$$
|b_i^{II}(s)| \leq \frac{\eta|S_i|\sqrt{n}(1 + R_g(m/\delta)^{1/d})^2\|\mathbf{f}(s) - \mathbf{y}\|_2}{\alpha^2 m}. \tag{C.10}
$$

The sum $\mathbf{f}(s+1) - \mathbf{f}(s) = \mathbf{a}^I(s) + \mathbf{a}^{II}(s) + \mathbf{b}^I(s) + \mathbf{b}^{II}(s)$ is separated into the primary term $\eta\mathbf{p}(s) = \mathbf{a}_I(s) + \mathbf{b}_I(s)$ and the residual term $\eta\mathbf{r}(s) = \mathbf{a}_{II}(s) + \mathbf{b}_{II}(s)$ which is a result of the discretization. With this, the evolution matrix $\mathbf{\Lambda}(s)$ in (C.5) is re-defined as

$$
\mathbf{\Lambda}(s) := \mathbf{G}(s) + \frac{\tilde{\mathbf{V}}(s) - \tilde{\mathbf{V}}^\perp(s)}{\alpha^2}
$$

and

$$
\mathbf{f}(s+1) - \mathbf{f}(s) = -\eta\mathbf{\Lambda}(s)(\mathbf{f}(s) - \mathbf{y}) + \eta\mathbf{r}(s).
$$

Now we compare $\|\mathbf{f}(s+1) - \mathbf{y}\|_2^2$ with $\|\mathbf{f}(s) - \mathbf{y}\|_2^2$,

$$
\begin{aligned}
\|\mathbf{f}(s+1) - \mathbf{y}\|_2^2 &= \|\mathbf{f}(s+1) - \mathbf{f}(s) + \mathbf{f}(s) - \mathbf{y}\|_2^2 \\
&= \|\mathbf{f}(s) - \mathbf{y}\|_2^2 + 2\langle \mathbf{f}(s+1) - \mathbf{f}(s), \ \mathbf{f}(s) - \mathbf{y} \rangle \\
&\quad + \langle \mathbf{f}(s+1) - \mathbf{f}(s), \ \mathbf{f}(s+1) - \mathbf{f}(s) \rangle.
\end{aligned}
$$

Substituting

$$
\mathbf{f}(s+1) - \mathbf{f}(s) = \mathbf{a}^I(s) + \mathbf{b}^I(s) + \mathbf{a}^{II}(s) + \mathbf{b}^{II}(s) = -\eta\mathbf{\Lambda}(s)(\mathbf{f}(s) - \mathbf{y}) + \eta\mathbf{r}(s)
$$

we obtain

$$
\begin{aligned}
\|\mathbf{f}(s+1) - \mathbf{y}\|_2^2 &= \|\mathbf{f}(s) - \mathbf{y}\|_2^2 + 2(-\eta\mathbf{\Lambda}(s)(\mathbf{f}(s) - \mathbf{y}) + \eta\mathbf{r}(s))^\top(\mathbf{f}(s) - \mathbf{y}) \\
&\quad + \eta^2(\mathbf{\Lambda}(s)(\mathbf{f}(s) - \mathbf{y}) - \mathbf{r}(s))^\top(\mathbf{\Lambda}(s)(\mathbf{f}(s) - \mathbf{y}) - \mathbf{r}(s)) \\
&\leq \|\mathbf{f}(s) - \mathbf{y}\|_2^2 + (\mathbf{f}(s) - \mathbf{y})^\top(-\eta\mathbf{\Lambda}(s) + \eta^2\mathbf{\Lambda}^2(s))(\mathbf{f}(s) - \mathbf{y}) \\
&\quad + \eta\mathbf{r}(s)^\top(\mathbf{I} - \eta\mathbf{\Lambda}(s))(\mathbf{f}(s) - \mathbf{y}) + \eta^2\|\mathbf{r}(s)\|_2^2.
\end{aligned}
$$

Now as $\lambda_{\min}(\mathbf{\Lambda}(s)) \geq \omega/2$ and $\eta = \frac{1}{3\|\mathbf{\Lambda}(s)\|_2}$, we have that

$$
(\mathbf{f}(s) - \mathbf{y})^\top(-\eta\mathbf{\Lambda}(s) + \eta^2\mathbf{\Lambda}^2(s))(\mathbf{f}(s) - \mathbf{y}) = -\eta(\mathbf{f}(s) - \mathbf{y})^\top(\mathbf{I} - \eta\mathbf{\Lambda}(s))\mathbf{\Lambda}(s)(\mathbf{f}(s) - \mathbf{y}) \leq -\frac{3\eta\omega}{8}\|\mathbf{f}(s) - \mathbf{y}\|_2^2.
$$

Next we analyze the rest of the terms and group them as $\mathbf{q}(s)$,

$$
\begin{aligned}
\mathbf{q}(s) &:= \eta\mathbf{r}(s)^\top(\mathbf{I} - \eta\mathbf{\Lambda}(s))(\mathbf{f}(s) - \mathbf{y}) + \eta^2\|\mathbf{r}(s)\|_2^2 \\
&\leq \eta\|\mathbf{r}(s)\|_2\|\mathbf{f}(s) - \mathbf{y}\|_2 + \eta^2\|\mathbf{r}(s)\|_2^2.
\end{aligned}
$$

By Property 1 we have

$$
\mathbf{q}(s) \leq \eta c\omega\|\mathbf{f}(s) - \mathbf{y}\|_2^2(1 + \eta c\omega) \leq 2c\eta\omega\|\mathbf{f}(s) - \mathbf{y}\|_2^2,
$$

so that

$$\mathbf{q}(s) \leq c' \eta \omega \|\mathbf{f}(s) - \mathbf{y}\|_2^2,$$

for $c'$ sufficiently small. Substituting, the difference $\mathbf{f}(s+1) - \mathbf{y}$ is bounded as

$$\|\mathbf{f}(s+1) - \mathbf{y}\|_2^2 \leq \|\mathbf{f}(s) - \mathbf{y}\|_2^2 - \eta \omega (1 - \eta \|\mathbf{\Lambda}(s)\|_2)\|\mathbf{f}(s) - \mathbf{y}\|_2^2 + c' \eta \omega \|\mathbf{f}(s) - \mathbf{y}\|_2^2$$
$$\leq (1 - \eta \omega (1 - \eta \|\mathbf{\Lambda}(s)\|_2) + c' \eta \omega)\|\mathbf{f}(s) - \mathbf{y}\|_2^2$$
$$\leq (1 - \eta \omega / 2)\|\mathbf{f}(s) - \mathbf{y}\|_2^2,$$

for well chosen absolute constant $c$. Hence for each $s = 0, 1, \ldots, K$,

$$\|\mathbf{f}(s+1) - \mathbf{y}\|_2^2 \leq (1 - \eta \omega / 2)\|\mathbf{f}(s) - \mathbf{y}\|_2^2$$

so the prediction error converges linearly. $\qquad \square$

In what comes next we prove the necessary conditions for Property 1, and define the appropriate $\omega$ for the $\mathbf{V}$ and $\mathbf{G}$ dominated regimes, in order to show $\lambda_{\min}(\mathbf{\Lambda}(s)) \geq \omega/2$.

**Proof of Theorem 4.1:**
To prove convergence we would like to apply Theorem C.1 with $\omega/2 = \frac{\lambda_0}{2\alpha^2}$. To do so we need to show that $m = \Omega(n^4 \log(n/\delta)/\lambda_0^4)$ guarantees that Property 1 holds and that $\lambda_{\min}(\mathbf{\Lambda}(s)) \geq \lambda_0/2\alpha^2$. For finite-step gradient training, take

$$R_v = \frac{\alpha \lambda_0}{192 n (m/\delta)^{1/d}}, \quad R_g = \frac{\lambda_0}{96 n (m/\delta)^{1/d}}. \tag{C.11}$$

Note the residual $\mathbf{r}(s)$ and the other terms $\mathbf{b}_I(s), \mathbf{b}_{II}(s)$ depend on the sets $S_i$ that we define here using $R_v$. We make the assumption that $\|\mathbf{v}_k(s) - \mathbf{v}_k(0)\|_2 \leq R_v$ and $|g_k(s) - g_k(0)| \leq R_g$ for all $k$ and that $s = 0, 1, \ldots K + 1$, this guarantees that $\mathbf{b}_I(s)$ and $\mathbf{\Lambda}(s)$ are well defined. Applying Lemmas B.1, B.4 with $R_v, R_g$ defined above, we have that $\lambda_{\min}(\tilde{\mathbf{V}}(s)) \geq \frac{5\lambda_0}{8}$. Then the least eigenvalue of the evolution matrix $\mathbf{\Lambda}(s)$ is bounded below

$$\lambda_{\min}(\mathbf{\Lambda}(s)) = \lambda_{\min}\left( \mathbf{G}(s) + \frac{\tilde{\mathbf{V}}(s) - \tilde{\mathbf{V}}^\perp(s)}{\alpha^2} \right)$$
$$\geq \lambda_{\min}\left( \frac{\tilde{\mathbf{V}}(s) - \tilde{\mathbf{V}}^\perp(s)}{\alpha^2} \right)$$
$$= \frac{\lambda_{\min}(\tilde{\mathbf{V}}(s) - \tilde{\mathbf{V}}^\perp(s))}{\alpha^2}$$
$$\geq \frac{5\lambda_0}{8\alpha^2} - \frac{\|\tilde{\mathbf{V}}^\perp(s)\|_2}{\alpha^2}.$$

The first inequality holds since $\mathbf{G}(s) \succ 0$ and the last inequality is since $\lambda_{\min}(\tilde{\mathbf{V}}(s)) \geq \frac{5\lambda_0}{8}$.

To show $\lambda_{\min}(\mathbf{\Lambda}(s)) \geq \frac{\lambda_0}{2\alpha^2}$ we bound $\|\tilde{\mathbf{V}}^\perp(s)\|_2 \leq \frac{\lambda_0}{8}$. By (C.9), we have

$$|\tilde{\mathbf{V}}_{ij}^\perp(s)| \leq \frac{(1 + R_g(m/\delta)^{1/d})|S_i|}{m} \leq (1 + R_g(m/\delta)^{1/d})\left( \frac{\sqrt{2}\tilde{R}_v}{\sqrt{\pi}\alpha} + \frac{16 \log(n/\delta)}{3m} \right).$$

Substituting $R_v, R_g$ and $m$, a direct calculation shows that

$$|\tilde{\mathbf{V}}_{ij}^\perp(s)| \leq \frac{\lambda_0}{8n},$$

which yields

$$\|\tilde{\mathbf{V}}^\perp(s)\|_2 \leq \|\tilde{\mathbf{V}}^\perp(s)\|_F \leq \frac{\lambda_0}{8}.$$

Hence $\lambda_{\min}(\mathbf{\Lambda}(s)) \geq \frac{\lambda_0}{2\alpha^2}$ for iterations $s = 0, 1, \ldots K$.

We proceed by showing the residual $\mathbf{r}(s)$ satisfies property 1. Recall $\mathbf{r}(s)$ is written as

$$\mathbf{r}(s) = \frac{\mathbf{a}^{II}(s)}{\eta} + \frac{\mathbf{b}^{II}(s)}{\eta}.$$

and Property 1 states that $\|\mathbf{r}(s)\|_2 \le \frac{c\eta\lambda_0}{\alpha^2}\|\mathbf{f}(s) - \mathbf{y}\|_2$ for sufficiently small absolute constant $c < 1$. This is equivalent to showing that both $\mathbf{a}^{II}(s)$, $\mathbf{b}^{II}(s)$ satisfy

$$\|\mathbf{a}^{II}(s)\|_2 \le \frac{c\eta\lambda_0}{\alpha^2}\|\mathbf{f}(s) - \mathbf{y}\|_2, \tag{C.12}$$

$$\|\mathbf{b}^{II}(s)\|_2 \le \frac{c\eta\lambda_0}{\alpha^2}\|\mathbf{f}(s) - \mathbf{y}\|_2. \tag{C.13}$$

We consider each term at turn. By (C.10),

$$\|\mathbf{b}^{II}(s)\|_2 \le \sqrt{n}\max_i b_i^{II}(s)$$

$$\le \max_i \frac{\eta n(1 + R_g(m/\delta)^{1/d})^2 |S_i| \|\mathbf{f}(s) - \mathbf{y}\|_2}{\alpha^2 m}$$

$$\le \frac{CmR_v \eta n \|\mathbf{f}(s) - \mathbf{y}\|_2}{\alpha^2 m}$$

$$\le \frac{\lambda_0 \eta \|\mathbf{f}(s) - \mathbf{y}\|_2}{\alpha^2} \cdot nCR_v.$$

In the above we used the values of $R_v$, $R_g$ defined in (C.11) and applied Lemma C.1 in the third inequality. Taking $m = \Omega\big(n^4 \log(n/\delta)/\lambda_0^4\big)$ with large enough constant yields

$$\|\mathbf{b}^{II}(s)\|_2 \le \frac{c\lambda_0 \eta \|\mathbf{f}(s) - \mathbf{y}\|_2}{\alpha^2}.$$

Next we analogously bound $\|\mathbf{a}^{II}(s)\|$ via the bound (C.7),

$$\|\mathbf{a}^{II}(s)\|_2 \le \sqrt{n}\max_i a_i^{II}(s)$$

$$\le \frac{\eta^2 n^{3/2}\big(1 + R_g(m/\delta)^{1/d}\big)^3 \|\mathbf{f}(s) - \mathbf{y}\|_2^2 (m/\delta)^{1/d}}{\alpha^4 \sqrt{m}}$$

$$\le \frac{\eta\lambda_0 \|\mathbf{f}(s) - \mathbf{y}\|_2}{\alpha^2} \cdot \frac{\eta\big(1 + R_g(m/\delta)^{1/d}\big)^3 n^{3/2}\|\mathbf{f}(s) - \mathbf{y}\|_2 (m/\delta)^{1/d}}{\lambda_0 \alpha^2 \sqrt{m}}$$

$$\le \frac{\eta\lambda_0 \|\mathbf{f}(s) - \mathbf{y}\|_2}{\alpha^2} \cdot \frac{\eta}{\alpha^2} \cdot \frac{Cn^2 \sqrt{\log(n/\delta)}}{\lambda_0 \sqrt{m}}$$

$$\le c\eta\omega \|\mathbf{f}(s) - \mathbf{y}\|_2.$$

In the above we applied Lemma B.9 once again. The last inequality holds since $m = \Omega(n^4 \log(n/\delta)/\lambda_0^4)$ and $\eta = O\Big(\frac{\alpha^2}{\|\mathbf{V}(s)\|_2}\Big)$, hence $\mathbf{r}(s)$ satisfies Property 1. Now since Theorem C.1 holds with $\omega = \lambda_0/\alpha^2$ we have that the maximum parameter trajectories are bounded as $\|\mathbf{v}_k(s) - \mathbf{v}_k(0)\|_2 \le R_v$ and $\|g_k(s) - g_k(0)\| \le R_g$ for all $k$ and every iteration $s = 0, 1, \ldots, K+1$ via Lemmas C.2, C.3.

To finish the proof, we apply the same contradiction argument as in Theorems B.1, B.2, taking the first iteration $s = K_0$ where one of Lemmas C.2, C.3 does not hold. We note that $K_0 > 0$ and by the definition of $K_0$, for $s = 0, 1, \ldots, K_0 - 1$ the Lemmas C.2, C.3 hold which implies that by the argument above we reach linear convergence in iteration $s = K_0$. This contradicts one of Lemmas C.2, C.3 which gives the desired contradiction, so we conclude that we have linear convergence with rate $\lambda_0/2\alpha^2$ for all iterations. $\qquad\square$

**Proof of Theorem 4.2:**
For $\mathbf{G}$-dominated convergence, we follow the same steps as in the proof of Theorem 4.1. We redefine

the trajectory constants for the finite step case

$$\tilde{R}_v := \frac{\sqrt{2\pi}\alpha\mu_0}{64n(m/\delta)^{1/d}}, \quad R_g := \frac{\mu_0}{48n(m/\delta)^{1/d}}.$$

To use Theorem C.1 we need to show that $m = \Omega\big(n^4\log(n/\delta)/\alpha^4\mu_0^4\big)$ guarantees Property 1, and that $\lambda_{\min}(\mathbf{\Lambda}(s)) \geq \mu_0/2$. We again note that the residual $\mathbf{r}(s)$ and $\mathbf{b}_I(s), \mathbf{b}_{II}(s)$ depend on the sets $S_i$ that we define here using $\tilde{R}_v$ above as $S_i := S_i(\tilde{R}_v)$.

We start by showing the property on the least eigenvalue. We make the assumption that we have linear convergence with $\omega/2 = \mu_0/2$ and step-size $\eta$ for iterations $s = 0, \ldots K$ so that Lemmas C.2, C.3 hold. Via an analogous analysis of the continous case we reach that $m = \Omega\big(n^4\log(n/\delta)/\mu_0^4\alpha^4\big)$ implies

$$\|\mathbf{v}_k(s) - \mathbf{v}_k(0)\|_2 \leq \frac{16\alpha\sqrt{n}\|\mathbf{f}(0) - \mathbf{y}\|_2}{\alpha\sqrt{m}\mu_0} \leq \tilde{R}_v, \quad |g_k(s) - g_k(0)| \leq \frac{8\sqrt{n}\|\mathbf{f}(0) - \mathbf{y}\|_2}{\sqrt{m}\mu_0} \leq R_g.$$

for $s = 0, \ldots K+1$ by Lemmas C.2, C.3 and that $\mathbf{\Lambda}(s), \mathbf{b}_I(s)$ are well defined. Using the bounds on the parameter trajectories, Lemma B.5 and $\tilde{R}_v$ defined above yield $\lambda_{\min}(\mathbf{G}(s)) \geq \frac{5\mu_0}{8}$. The least eigenvalue of the evolution matrix $\mathbf{\Lambda}(s)$ is bounded below as

$$\lambda_{\min}(\mathbf{\Lambda}(s)) = \lambda_{\min}\left(\mathbf{G}(s) + \frac{\tilde{\mathbf{V}}(s) - \tilde{\mathbf{V}}^\perp(s)}{\alpha^2}\right)$$

$$\geq \lambda_{\min}(\mathbf{G}(s)) - \|\tilde{\mathbf{V}}^\perp(s)\|_2$$

since $\tilde{\mathbf{V}}(s) \succ 0$ and $\alpha \geq 1$. We bound the spectral norm of $\|\tilde{\mathbf{V}}^\perp(s)\|_2$, for each entry $i, j$ we have by (C.9) that

$$|\tilde{\mathbf{V}}_{ij}^\perp(s)| \leq \frac{(1 + R_g(m/\delta)^{1/d})|S_i|}{m}$$

$$\leq (1 + R_g(m/\delta)^{1/d})\left(\frac{\sqrt{2}\tilde{R}_v}{\sqrt{\pi}\alpha} + \frac{16\log(n/\delta)}{3m}\right)$$

$$\leq \frac{8\tilde{R}_v}{\sqrt{2\pi}\alpha}$$

$$\leq \frac{\mu_0}{8n}.$$

where in the above inequalities we used our bounds on $\tilde{R}_v, R_g$ and $m$. Then the spectral norm is bounded as

$$\|\tilde{\mathbf{V}}^\perp(s)\|_2 \leq \|\tilde{\mathbf{V}}^\perp(s)\|_F \leq \mu_0/8.$$

Hence we have that $\lambda_{\min}(\mathbf{\Lambda}(s)) \geq \mu_0/2$ for $s = 0, 1, \ldots K$.

Next we show the residual $\mathbf{r}(s)$ satisfies Property 1. Recall $\mathbf{r}(s)$ is written as

$$\mathbf{r}(s) = \frac{\mathbf{a}^{II}(s)}{\eta} + \frac{\mathbf{b}^{II}(s)}{\eta}.$$

Property 1 states the condition $\|\mathbf{r}(s)\|_2 \leq c\omega\eta\|\mathbf{f}(s) - \mathbf{y}\|_2$ for sufficiently small $c < 1$ with $\omega = \mu_0$. This is equivalent to showing that both $\mathbf{a}^{II}(s), \mathbf{b}^{II}(s)$ satisfy that

$$\|\mathbf{a}^{II}(s)\|_2 \leq c\eta\mu_0\|\mathbf{f}(s) - \mathbf{y}\|_2, \tag{C.14}$$

$$\|\mathbf{b}^{II}(s)\|_2 \leq c\eta\mu_0\|\mathbf{f}(s) - \mathbf{y}\|_2, \tag{C.15}$$

for sufficiently small absolute constant $c$. For $\mathbf{b}_{II}(s)$ we have that (C.10) gives

$$\|\mathbf{b}^{II}(s)\|_2 \leq \sqrt{n}\max_i b_i^{II}(s)$$

$$\leq \max_i \frac{\eta(1 + R_g(m/\delta)^{1/d})^2|S_i|n\|\mathbf{f}(s) - \mathbf{y}\|_2}{\alpha^2 m}.$$

Next applying Lemmas C.1 and B.9 in turn yields

$$\leq \frac{Cm\tilde{R}_v\eta n\|\mathbf{f}(s) - \mathbf{y}\|_2}{\alpha^2 m}$$

$$\leq \eta\mu_0\|\mathbf{f}(s) - \mathbf{y}\|_2 \frac{\tilde{R}_v}{n\alpha^2}$$

Substituting $m = \Omega\big(n^4 \log(n/\delta)/\mu_0^4\alpha^4\big)$ for a large enough constant and $R_v$ we get

$$\|\mathbf{b}^{II}(s)\|_2 \leq c\eta\mu_0\|\mathbf{f}(s) - \mathbf{y}\|_2.$$

Analogously we bound $\|\mathbf{a}^{II}(s)\|_2$ using (C.7),

$$\|\mathbf{a}^{II}(s)\|_2 \leq \sqrt{n} \max_i a_i^{II}(s)$$

$$\leq \frac{\eta^2 n^{3/2}\big(1 + R_g(m/\delta)^{1/d}\big)^3\|\mathbf{f}(s) - \mathbf{y}\|_2^2(m/\delta)^{1/d}}{\alpha^4\sqrt{m}}$$

$$\leq \eta\mu_0\|\mathbf{f}(s) - \mathbf{y}\|_2 \cdot \frac{\eta\big(1 + R_g(m/\delta)^{1/d}\big)^3 n^{3/2}\|\mathbf{f}(s) - \mathbf{y}\|_2(m/\delta)^{1/d}}{\mu_0\alpha^4\sqrt{m}}$$

$$\leq \eta\mu_0\|\mathbf{f}(s) - \mathbf{y}\|_2 \cdot \frac{\eta}{\alpha^2} \cdot \frac{Cn^2\sqrt{\log(n/\delta)}}{\alpha^2\mu_0^2\sqrt{m}}$$

$$\leq c\eta\mu_0\|\mathbf{f}(s) - \mathbf{y}\|_2.$$

Where we have used Lemma B.9 in the third inequality and substituted $m = \Omega(n^4 \log(n/\delta)/\alpha^4\mu_0^4)$ noting that $\eta = O\big(\frac{1}{\|\mathbf{\Lambda}(s)\|_2}\big)$ and that $\alpha \geq 1$ in the last inequality. Therefore we have that $\mathbf{r}(s)$ satisfies Property 1 so that Theorem C.1 holds. By the same contradiction argument as in Theorem 4.1 we have that this holds for all iterations. □

# D  ADDITIONAL TECHNICAL LEMMAS AND PROOFS OF LEMMAS FROM APPENDIX B

**Proof of Lemma 4.1:**
We prove Lemma 4.1 for $\mathbf{V}^\infty$, $\mathbf{G}^\infty$ separately. $\mathbf{V}^\infty$ can be viewed as the covariance matrix of the functionals $\phi_i$ defined as

$$\phi_i(\mathbf{v}) = \mathbf{x}_i\left(\mathbf{I} - \frac{\mathbf{v}\mathbf{v}^\top}{\|\mathbf{v}\|_2^2}\right)\mathbb{1}\{\mathbf{v}^\top\mathbf{x}_i \geq 0\} \tag{D.1}$$

over the Hilbert space $\mathcal{V}$ of $L^2(N(0, \alpha^2\mathbf{I}))$ of functionals. Under this formulation, if $\phi_1, \phi_2, \ldots, \phi_n$ are linearly independent, then $\mathbf{V}^\infty$ is strictly positive definite. Thus, to show that $\mathbf{V}^\infty$ is strictly positive definite is equivalent to showing that

$$c_1\phi_1 + c_2\phi_2 + \cdots + c_n\phi_n = 0 \ \text{ in } \mathcal{V} \tag{D.2}$$

implies $c_i = 0$ for each $i$. The $\phi_i$s are piece-wise continuous functionals, and equality in $\mathcal{V}$ is equivalent to

$$c_1\phi_1 + c_2\phi_2 + \cdots + c_n\phi_n = 0 \ \text{ almost everywhere.}$$

For the sake of contradiction, assume that there exist $c_1, \ldots, c_n$ that are not identically 0, satisfying (D.2). As $c_i$ are not identically 0, there exists an $i$ such that $c_i \neq 0$. We show this leads to a contradiction by constructing a non-zero measure region such that the linear combination $\sum_i c_i\phi_i$ is non-zero.

Denote the orthogonal subspace to $\mathbf{x}_i$ as $D_i := \{\mathbf{v} \in \mathbb{R}^d : \mathbf{v}^\top\mathbf{x}_i = 0\}$. By Assumption 1,

$$D_i \not\subseteq \bigcup_{j\neq i} D_j$$

This holds since $D_i$ is a $(d-1)$-dimensional space which may not be written as the finite union of subspaces $D_i \cap D_j$ of dimension $d-2$ (since $\mathbf{x}_i$ and $\mathbf{x}_j$ are not parallel). Thus, take $\mathbf{z} \in D_i \backslash \bigcup_{j \neq i} D_j$. Since $\bigcup_{j \neq i} D_j$ is closed in $\mathbb{R}^d$, there exists an $R > 0$ such that

$$\mathcal{B}(\mathbf{z}, 4R) \cap \bigcup_{j \neq i} D_j = \emptyset.$$

Next take $\mathbf{y} \in \partial \mathcal{B}(\mathbf{z}, 3R) \cap D_i$ (where $\partial$ denotes the boundary) on the smaller disk of radius $3R$ so that it satisfies $\|\mathbf{y}\|_2 = \max_{\mathbf{y}' \in \partial \mathcal{B}(\mathbf{z}, 3R) \cap D_i} \|\mathbf{y}'\|_2$. Now for any $r \leq R$, the ball $\mathcal{B}(\mathbf{y}, r)$ is such that for all points $\mathbf{v} \in \mathcal{B}(\mathbf{y}, r)$ we have $\|\mathbf{v}^{\mathbf{x}_i^\perp}\|_2 \geq 2R$ and $\|\mathbf{v}^{\mathbf{x}_i}\|_2 \leq R$. Then for any $r \leq R$, the points $\mathbf{v} \in \mathcal{B}(\mathbf{y}, r) \subset \mathcal{B}(\mathbf{z}, 4R)$ satisfy that

$$\|\mathbf{x}_i^{\mathbf{v}^\perp}\|_2 \geq \|\mathbf{x}_i\|_2 - \frac{\mathbf{x}_i \cdot \mathbf{v}}{\|\mathbf{v}\|_2} \geq \|\mathbf{x}_i\|_2 \left(1 - \frac{R}{2R}\right) \geq \frac{\|\mathbf{x}_i\|_2}{2}.$$

Next we decompose the chosen ball $\mathcal{B}(\mathbf{y}, r) = B^+(r) \vee B^-(r)$ to the areas where the ReLU function at the point $\mathbf{x}_i$ is active and inactive

$$B^+(r) = \mathcal{B}(\mathbf{y}, r) \cap \{\mathbf{x}_i^\top \mathbf{v} \geq 0\}, \quad B^-(r) = \mathcal{B}(\mathbf{y}, r) \cap \{\mathbf{x}_i^\top \mathbf{v} < 0\}.$$

Note that $\phi_i$ has a discontinuity on $D_i$ and is continuous within each region $B^+(r)$ and $B^-(r)$. Moreover, for $j \neq i$, $\phi_j$ is continuous on the entire region of $\mathcal{B}(\mathbf{y}, r)$ since $\mathcal{B}(\mathbf{y}, r) \subset \mathcal{B}(\mathbf{z}, 4R) \subset D_j^c$. Since we have that $\phi_j$ is continuous in the region, the Lebesgue differentiation theorem implies that for $r \to 0$, $\phi_i$ satisfies on $B^+(r), B^-(r)$:

$$\lim_{r \to 0} \frac{1}{\mu(B^+(r))} \int_{B^+(r)} \phi_i = \mathbf{x}_i^{\mathbf{y}^\perp} \neq 0, \quad \lim_{r \to 0} \frac{1}{\mu(B^-(r))} \int_{B^-(r)} \phi_i = 0.$$

For $j \neq i$ $\phi_j$ is continuous on the entire ball $\mathcal{B}(\mathbf{y}, r)$ hence the Lebesgue differentiation theorem also gives

$$\lim_{r \to 0} \frac{1}{\mu(B^+(r))} \int_{B^+(r)} \phi_i = \phi_j(\mathbf{y}), \quad \lim_{r \to 0} \frac{1}{\mu(B^-(r))} \int_{B^-(r)} \phi_i = \phi_j(\mathbf{y}).$$

We integrate $c_1 \phi_1 + \ldots c_n \phi_n$ over $B^-(r)$ and $B^+(r)$ separately and subtract the integrals. By the assumption, $c_1 \phi_1 + \cdots + c_n \phi_n = 0$ almost everywhere so each integral evaluates to 0 and the difference is also 0,

$$0 = \frac{1}{\mu(B^+(r))} \int_{B^+(r)} c_1 \phi_1 + \cdots + c_n \phi_n - \frac{1}{\mu(B^-(r))} \int_{B^-(r)} c_1 \phi_1 + \cdots + c_n \phi_n. \quad \text{(D.3)}$$

By the continuity of $\phi_j$ for $j \neq i$ taking $r \to 0$ we have that

$$\frac{1}{\mu(B^+(r))} \lim_{r \to 0} \int_{B^+(r)} \phi_j - \frac{1}{\mu(B^-(r))} \int_{B^-(r)} \phi_j = \phi_j(\mathbf{y}) - \phi_j(\mathbf{y}) = 0.$$

For $\phi_i$ the functionals evaluate differently. For $B^-(r)$ we have that

$$\frac{1}{\mu(B^-(r))} \lim_{r \to 0} \int_{B^-(r)} \phi_i = \frac{1}{\mu(B^-(r))} \lim_{r \to 0} \int_{B^-(r)} 0 = 0,$$

while the integral over the positive side, $B^+(r)$ is equal to

$$\frac{1}{\mu(B^+(r))} \int_{B^+(r)} \phi_i(\mathbf{z}) d\mathbf{z} = \frac{1}{\mu(B^+(r))} \int_{B^+(r)} \mathbf{x}_i^{\mathbf{z}^\perp} d\mathbf{z} = \mathbf{x}_i^{\mathbf{y}^\perp}.$$

By construction, $\|\mathbf{x}_i^{\mathbf{y}^\perp}\|_2 > R$ and is non-zero so we conclude that for (D.3) to hold we must have $c_i = 0$. This gives the desired contradiction and implies that $\phi_1, \ldots \phi_n$ are independent and $\mathbf{V}^\infty$ is positive definite with $\lambda_{\min}(\mathbf{V}^\infty) = \lambda_0$.

Next we consider $\mathbf{G}^\infty$ and again frame the problem in the context of the covariance matrix of functionals. Define

$$\theta_i(\mathbf{v}) := \sigma\left(\frac{\mathbf{v}^\top \mathbf{x}_i}{\|\mathbf{v}\|_2}\right)$$

for $\mathbf{v} \neq 0$. Now the statement of the theorem is equivalent to showing that the covariance matrix of $\{\theta_i\}$ does not have zero-eigenvalues, that is, the functionals $\theta_i$s are linearly independent. For the sake of contradiction assume $\exists\, c_1, \ldots, c_n$ such that

$$c_1\theta_1 + c_2\theta_2 + \cdots + c_n\theta_n = 0 \quad \text{in } \mathcal{V} \quad \text{(equivalent to a.e).}$$

Via the same contradiction argument we show that $c_i = 0$ for all $i$. Unlike $\phi_i$ defined in (D.1), each $\theta_i$ is continuous and non-negative so equality "a.e" is strengthened to "for all $\mathbf{v}$",

$$c_1\theta_1 + c_2\theta_2 + \cdots + c_n\theta_n = 0.$$

Equality everywhere requires that the derivatives of the function are equal to $0$ almost everywhere. Computing derivatives with respect to $\mathbf{v}$ yields

$$c_1 \mathbf{x}_1^{\mathbf{v}^\perp} \mathbb{1}\{\mathbf{v}^\top \mathbf{x}_1 \geq 0\} + c_2 \mathbf{x}_2^{\mathbf{v}^\perp} \mathbb{1}\{\mathbf{v}^\top \mathbf{x}_2 \geq 0\} + \cdots + c_n \mathbf{x}_n^{\mathbf{v}^\perp} \mathbb{1}\{\mathbf{v}^\top \mathbf{x}_n \geq 0\} = 0.$$

Which coincide with

$$c_1\phi_1 + \cdots + c_n\phi_n$$

By the first part of the proof, the linear combination $c_1\phi_1 + \cdots + c_n\phi_n$ is non-zero around a ball of positive measure unless $c_i = 0$ for all $i$. This contradicts the assumption that the derivative is $0$ almost everywhere; therefore $\mathbf{G}^\infty$ is strictly positive definite with $\lambda_{\min}(G^\infty) =: \mu_0 > 0$. □

We briefly derive an inequality for the sum of indicator functions for events that are bounded by the sum of indicator functions of *independent* events. This enables us to develop more refined concentration than in Du et al. [15] for monitoring the orthogonal and aligned Gram matrices during training.

**Lemma D.1.** *Let $A_1, \ldots, A_m$ be a sequence of events and suppose that $A_k \subseteq B_k$ with $B_1, \ldots, B_m$ mutually independent. Further assume that for each $k$, $\mathbb{P}(B_k) \leq p$, and define $S = \frac{1}{m}\sum_{k=1}^m \mathbb{1}_{A_k}$. Then with probability $1 - \delta$, $S$ satisfies*

$$S \leq p\left(2 + \frac{8\log(1/\delta)}{3mp}\right).$$

**Proof of Lemma D.1:**
Bound $S$ as

$$S = \frac{1}{m}\sum_{k=1}^m \mathbb{1}_{A_k} \leq \frac{1}{m}\sum_{k=1}^m \mathbb{1}_{B_k}.$$

We apply Bernstein's concentration inequality to reach the bound. Denote $X_k = \frac{\mathbb{1}_{B_k}}{m}$ and $\tilde{S} = \sum_{k=1}^m X_k$. Then

$$\text{Var}(X_k) \leq \mathbb{E}X_k^2 = (1/m)^2 \mathbb{P}(X_k) + 0 \leq \frac{p}{m^2}, \quad \mathbb{E}\tilde{S} = \mathbb{E}\sum_{k=1}^m X_k \leq p.$$

Applying Bernstein's inequality yields

$$\mathbb{P}(\tilde{S} - \mathbb{E}\tilde{S} \geq t) \leq \exp\left(\frac{-t^2/2}{\sum_{k=1}^m \mathbb{E}X_k^2 + \frac{t}{3m}}\right).$$

Fix $\delta$ and take the smallest $t$ such that $\mathbb{P}(\tilde{S} - \mathbb{E}\tilde{S} \geq t) \leq \delta$. Denote $t = r \cdot \mathbb{E}\tilde{S}$, either $\mathbb{P}(\tilde{S} - \mathbb{E}\tilde{S} \geq \mathbb{E}\tilde{S}) \leq \delta$, or $t = r\mathbb{E}\tilde{S}$ corresponds to $r \geq 1$. Note that $t = r\mathbb{E}\tilde{S} \leq rp$. In the latter case, the bound is written as

$$\mathbb{P}(\tilde{S} - \mathbb{E}\tilde{S} \geq rp) \leq \exp\left(\frac{-(pr)^2/2}{p/m + \frac{pr}{3m}}\right) \leq \exp\left(\frac{-(pr)^2/2}{\frac{p}{m}(1 + \frac{r}{3})}\right) \leq \exp\left(\frac{-(pr)^2/2}{\frac{p}{m}(\frac{4r}{3})}\right) = \exp\left(\frac{-3prm}{8}\right).$$

Solving for $\delta$ gives

$$rp \leq \frac{8\log(1/\delta)}{3m}.$$

Hence with probability $1 - \delta$,

$$S \leq \tilde{S} \leq \max\left\{ p\left(1 + \frac{8\log(1/\delta)}{3mp}\right), 2p \right\} \leq p\left(2 + \frac{8\log(1/\delta)}{3mp}\right).$$

$\square$

**Proof of Lemma B.1:**
We prove the claim by applying concentration on each entry of the difference matrix. Each entry $\mathbf{V}_{ij}(0)$ is written as

$$\mathbf{V}_{ij}(0) = \frac{1}{m}\sum_{k=1}^{m} \left\langle \mathbf{x}_i^{\mathbf{v}_k(0)^\perp}, \mathbf{x}_j^{\mathbf{v}_k(0)^\perp} \right\rangle \left(\frac{\alpha c_k \cdot g_k}{\|\mathbf{v}_k\|_2}\right)^2 \mathbb{1}_{ik}(0)\mathbb{1}_{jk}(0).$$

At initialization $g_k(0) = \|\mathbf{v}_k(0)\|_2/\alpha$, $c_k^2 = 1$ so $\mathbf{V}_{ij}(0)$ simplifies to

$$\mathbf{V}_{ij}(0) = \frac{1}{m}\sum_{k=1}^{m} \left\langle \mathbf{x}_i^{\mathbf{v}_k(0)^\perp}, \mathbf{x}_j^{\mathbf{v}_k(0)^\perp} \right\rangle \mathbb{1}_{ik}(0)\mathbb{1}_{jk}(0).$$

Since the weights $\mathbf{v}_k(0)$ are initialized independently for each entry we have $\mathbb{E}_{\mathbf{v}}\mathbf{V}_{ij}(0) = \mathbf{V}_{ij}^\infty$. We measure the deviation $\mathbf{V}(0) - \mathbf{V}^\infty$ via concentration. Each term in the sum $\frac{1}{m}\sum_{j=1}^{m} \left\langle \mathbf{x}_i^{\mathbf{v}_k(0)^\perp}, \mathbf{x}_j^{\mathbf{v}_k(0)^\perp} \right\rangle \mathbb{1}_{ik}(0)\mathbb{1}_{jk}(0)$ is independent and bounded,

$$-1 \leq \left\langle \mathbf{x}_i^{\mathbf{v}_k(0)^\perp}, \mathbf{x}_j^{\mathbf{v}_k(0)^\perp} \right\rangle \mathbb{1}_{ik}(0)\mathbb{1}_{jk}(0) \leq 1.$$

Applying Hoeffding's inequality to each entry yields that with probability $1 - \delta/n^2$, for all $i, j$,

$$|\mathbf{V}_{ij}(0) - \mathbf{V}_{ij}^\infty| \leq \frac{2\sqrt{\log(n^2/\delta)}}{\sqrt{m}}.$$

Taking a union bound over all entries, with probability $1 - \delta$,

$$|\mathbf{V}_{ij}(0) - \mathbf{V}_{ij}^\infty| \leq \frac{4\sqrt{\log(n/\delta)}}{\sqrt{m}}.$$

Bounding the spectral norm, with probability $1 - \delta$,

$$\|\mathbf{V}(0) - \mathbf{V}^\infty\|_2^2 \leq \|\mathbf{V}(0) - \mathbf{V}^\infty\|_F^2 \leq \sum_{i,j} |\mathbf{V}_{ij}(0) - \mathbf{V}_{ij}^\infty|^2$$

$$\leq \frac{16n^2\log(n/\delta)}{m}.$$

Taking $m = \Omega\left(\frac{n^2\log(n/\delta)}{\lambda_0^2}\right)$ therefore guarantees

$$\|\mathbf{V}(0) - \mathbf{V}^\infty\|_2 \leq \frac{\lambda_0}{4}.$$

$\square$

**Proof of Lemma B.2:**
This is completely analogous to B.1. Recall $\mathbf{G}(0)$ is defined as,

$$\mathbf{G}_{ij}(0) = \frac{1}{m}\sum_{k=1}^{m} \left\langle \mathbf{x}_i^{\mathbf{v}_k(0)}, \mathbf{x}_j^{\mathbf{v}_k(0)} \right\rangle c_k^2 \mathbb{1}_{ik}(0)\mathbb{1}_{jk}(0)$$

with $c_k^2 = 1$ and $\mathbf{v}_k(0) \sim N(0, \alpha^2 \mathbf{I})$ are initialized i.i.d. Since each term is bounded like B.1 the same analysis gives

$$\|\mathbf{G}_{ij}(0) - \mathbf{G}_{ij}^\infty\|_2^2 \leq \frac{16n^2 \log(n/\delta)}{m}.$$

Taking $m = \Omega\left(\frac{n^2 \log(n/\delta)}{\mu_0^2}\right)$ therefore guarantees,

$$\|\mathbf{G}(0) - \mathbf{G}^\infty\|_2 \leq \frac{\mu_0}{4}.$$

$\square$

**Proof of Lemma B.3:**

For a given $R$, define the event of a possible sign change of neuron $k$ at point $\mathbf{x}_i$ as

$$A_{i,k}(R) = \{\exists \mathbf{v} : \|\mathbf{v} - \mathbf{v}_k(0)\|_2 \leq R, \text{ and } \mathbb{1}\{\mathbf{v}_k(0)^\top \mathbf{x}_i \geq 0\} \neq \mathbb{1}\{\mathbf{v}^\top \mathbf{x}_i \geq 0\}\}$$

$A_{i,k}(R)$ occurs exactly when $|\mathbf{v}_k(0)^\top \mathbf{x}_i| \leq R$, since $\|\mathbf{x}_i\|_2 = 1$ and the perturbation may be taken in the direction of $-\mathbf{x}_i$. To bound the probability $A_{i,k}(R)$ we consider the probability of the event

$$\mathbb{P}(A_{i,k}(R)) = \mathbb{P}(|\mathbf{v}_k(0)^\top \mathbf{x}_i| < R) = \mathbb{P}(|z| < R).$$

Here, $z \sim N(0, \alpha^2)$ since the product $\mathbf{v}_k(0)^\top \mathbf{x}_i$ follows a centered normal distribution. The norm of $\|\mathbf{x}_i\|_2 = 1$ which implies that $z$ computes to a standard deviation $\alpha$. Via estimates on the normal distribution, the probability on the event is bounded like

$$\mathbb{P}(A_{i,k}(R)) \leq \frac{2R}{\alpha\sqrt{2\pi}}.$$

We use the estimate for $\mathbb{P}(A_{i,k}(R))$ to bound the difference between the surrogate Gram matrix and the Gram matrix at initialization $\mathbf{V}(0)$.

Recall the surrogate $\hat{\mathbf{V}}(t)$ is defined as,

$$\hat{\mathbf{V}}_{ij}(t) = \frac{1}{m} \sum_{k=1}^m \langle \mathbf{x}_i^{\mathbf{v}_k(t)^\perp}, \mathbf{x}_k^{\mathbf{v}_k(t)^\perp} \rangle \mathbb{1}_{ik}(t)\mathbb{1}_{jk}(t).$$

Thus for entry $i, j$ we have

$$|\hat{\mathbf{V}}_{ij}(t) - \mathbf{V}_{ij}(0)| = \left| \frac{1}{m} \sum_{k=1}^m \langle \mathbf{x}_i^{\mathbf{v}_k(t)^\perp}, \mathbf{x}_j^{\mathbf{v}_k(t)^\perp} \rangle \mathbb{1}_{ik}(t)\mathbb{1}_{jk}(t) - \langle \mathbf{x}_i^{\mathbf{v}_k(0)^\perp}, \mathbf{x}_j^{\mathbf{v}_k(0)^\perp} \rangle \mathbb{1}_{ik}(0)\mathbb{1}_{jk}(0) \right|$$

This sum is decomposed into the difference between the inner product and the difference in the rectifier patterns terms respectively:

$$\left( \langle \mathbf{x}_i^{\mathbf{v}_k(t)^\perp}, \mathbf{x}_j^{\mathbf{v}_k(t)^\perp} \rangle - \langle \mathbf{x}_i^{\mathbf{v}_k(0)^\perp}, \mathbf{x}_j^{\mathbf{v}_k(0)^\perp} \rangle \right), \qquad \left( \mathbb{1}_{ik}(t)\mathbb{1}_{jk}(t) - \mathbb{1}_{ik}(0)\mathbb{1}_{jk}(0) \right).$$

Define

$$Y_{ij}^k = \left( \langle \mathbf{x}_i^{\mathbf{v}_k(t)^\perp}, \mathbf{x}_j^{\mathbf{v}_k(t)^\perp} \rangle - \langle \mathbf{x}_i^{\mathbf{v}_k(0)^\perp}, \mathbf{x}_j^{\mathbf{v}_k(0)^\perp} \rangle \right) \left( \mathbb{1}_{ik}(t)\mathbb{1}_{jk}(t) \right),$$

$$Z_{ij}^k = \left( \langle \mathbf{x}_i^{\mathbf{v}_k(0)^\perp}, \mathbf{x}_j^{\mathbf{v}_k(0)^\perp} \rangle \right) \left( \mathbb{1}_{ik}(t)\mathbb{1}_{jk}(t) - \mathbb{1}_{ik}(0)\mathbb{1}_{jk}(0) \right).$$

Then

$$|\hat{\mathbf{V}}_{ij}(t) - \mathbf{V}_{ij}(0)| = \left| \frac{1}{m} \sum_{k=1}^m Y_{ij}^k + Z_{ij}^k \right| \leq \left| \frac{1}{m} \sum_{k=1}^m Y_{ij}^k \right| + \left| \frac{1}{m} \sum_{k=1}^m Z_{ij}^k \right|.$$

To bound $|\frac{1}{m} \sum_{k=1}^m Y_{ij}^k|$ we bound each $|Y_{ij}^k|$ as follows.

$$
\begin{aligned}
|Y_{ij}^k| &= \left| \left( \left\langle \mathbf{x}_i^{\mathbf{v}_k(t)^\perp}, \mathbf{x}_j^{\mathbf{v}_k(t)^\perp} \right\rangle - \left\langle \mathbf{x}_i^{\mathbf{v}_k(0)^\perp}, \mathbf{x}_j^{\mathbf{v}_k(0)^\perp} \right\rangle \right) \left( \mathbb{1}_{ik}(t) \mathbb{1}_{jk}(t) \right) \right| \\
&\le \left| \left\langle \mathbf{x}_i^{\mathbf{v}_k(t)^\perp}, \mathbf{x}_j^{\mathbf{v}_k(t)^\perp} \right\rangle - \left\langle \mathbf{x}_i^{\mathbf{v}_k(0)^\perp}, \mathbf{x}_j^{\mathbf{v}_k(0)^\perp} \right\rangle \right| \\
&= \left| \left\langle \mathbf{x}_i, \mathbf{x}_j \right\rangle - \left\langle \mathbf{x}_i^{\mathbf{v}_k(t)}, \mathbf{x}_j^{\mathbf{v}_k(t)} \right\rangle + \left\langle \mathbf{x}_i^{\mathbf{v}_k(0)}, \mathbf{x}_j^{\mathbf{v}_k(0)} \right\rangle - \left\langle \mathbf{x}_i, \mathbf{x}_j \right\rangle \right| \\
&= \left| \left\langle \frac{\mathbf{x}_i^\top \mathbf{v}_k(t)}{\|\mathbf{v}_k(t)\|_2} \cdot \frac{\mathbf{v}_k(t)}{\|\mathbf{v}_k(t)\|_2}, \frac{\mathbf{x}_j^\top \mathbf{v}_k(t)}{\|\mathbf{v}_k(t)\|_2} \cdot \frac{\mathbf{v}_k(t)}{\|\mathbf{v}_k(t)\|_2} \right\rangle - \left\langle \mathbf{x}_i^{\mathbf{v}_k(0)}, \mathbf{x}_j^{\mathbf{v}_k(0)} \right\rangle \right| \\
&= \left| \frac{\mathbf{x}_i^\top \mathbf{v}_k(t)}{\|\mathbf{v}_k(t)\|_2} \cdot \frac{\mathbf{x}_j^\top \mathbf{v}_k(t)}{\|\mathbf{v}_k(t)\|_2} - \left\langle \mathbf{x}_i^{\mathbf{v}_k(0)}, \mathbf{x}_j^{\mathbf{v}_k(0)} \right\rangle \right| \\
&= \left| \frac{\mathbf{x}_i^\top \mathbf{v}_k(0)}{\|\mathbf{v}_k(0)\|_2} \cdot \frac{\mathbf{x}_j^\top \mathbf{v}_k(0)}{\|\mathbf{v}_k(0)\|_2} + \mathbf{x}_i^\top \left( \frac{\mathbf{v}_k(t)}{\|\mathbf{v}_k(t)\|_2} - \frac{\mathbf{v}_k(0)}{\|\mathbf{v}_k(0)\|_2} \right) \cdot \frac{\mathbf{x}_j^\top \mathbf{v}_k(t)}{\|\mathbf{v}_k(t)\|_2} \right. \\
&\qquad \left. + \mathbf{x}_j^\top \left( \frac{\mathbf{v}_k(t)}{\|\mathbf{v}_k(t)\|_2} - \frac{\mathbf{v}_k(0)}{\|\mathbf{v}_k(0)\|_2} \right) \cdot \frac{\mathbf{x}_i^\top \mathbf{v}_k(0)}{\|\mathbf{v}_k(0)\|_2} - \left\langle \mathbf{x}_i^{\mathbf{v}_k(0)}, \mathbf{x}_j^{\mathbf{v}_k(0)} \right\rangle \right| \\
&\le \left| \mathbf{x}_i^\top \left( \frac{\mathbf{v}_k(t)}{\|\mathbf{v}_k(t)\|_2} - \frac{\mathbf{v}_k(0)}{\|\mathbf{v}_k(0)\|_2} \right) \cdot \frac{\mathbf{x}_j^\top \mathbf{v}_k(t)}{\|\mathbf{v}_k(t)\|_2} \right| + \left| \mathbf{x}_i^\top \left( \frac{\mathbf{v}_k(t)}{\|\mathbf{v}_k(t)\|_2} - \frac{\mathbf{v}_k(0)}{\|\mathbf{v}_k(0)\|_2} \right) \cdot \frac{\mathbf{x}_j^\top \mathbf{v}_k(t)}{\|\mathbf{v}_k(t)\|_2} \right| \\
&\le 2 \left\| \frac{\mathbf{v}_k(t)}{\|\mathbf{v}_k(t)\|_2} - \frac{\mathbf{v}_k(0)}{\|\mathbf{v}_k(0)\|_2} \right\|_2.
\end{aligned}
$$

Therefore, we have

$$
\begin{aligned}
\left| \frac{1}{m} \sum_{k=1}^m Y_{ij}^k \right| &\le \frac{2}{m} \sum_{k=1}^m \left\| \frac{\mathbf{v}_k(t)}{\|\mathbf{v}_k(t)\|_2} - \frac{\mathbf{v}_k(0)}{\|\mathbf{v}_k(0)\|_2} \right\|_2 \\
&\le \frac{4 R_{\mathbf{v}} (2m/\delta)^{1/d}}{\alpha} \\
&\le \frac{8 R_{\mathbf{v}} (m/\delta)^{1/d}}{\alpha},
\end{aligned}
$$

where the first inequality follows from Lemma B.10. Note that the inequality holds with high probability $1 - \delta/2$ for all $i, j$.

For the second sum, $\left| \frac{1}{m} \sum_{k=1}^m Z_{ij}^k \right| \le \frac{1}{m} \sum_{k=1}^m \mathbb{1}_{A_{ik}(R)} + \frac{1}{m} \sum_{k=1}^m \mathbb{1}_{A_{jk}(R)}$ so we apply Lemma D.1 to get, with probability $1 - \delta/2n^2$

$$
\begin{aligned}
\left| \frac{1}{m} \sum_{k=1}^m Z_{ij}^k \right| &\le \frac{2 R_v}{\alpha \sqrt{2\pi}} \left( 2 + \frac{2\sqrt{2\pi} \alpha \log (2n^2/\delta)}{3 m R_v} \right) \\
&\le \frac{8 R_v}{\alpha \sqrt{2\pi}}
\end{aligned}
$$

since $m$ satisfies $m = \Omega\left( \frac{(m/\delta)^{1/d} n^2 \log(n/\delta)}{\alpha \lambda_0} \right)$. Combining the two sums for $Y_{ij}^k$ and $Z_{ij}^k$, with probability $1 - \frac{\delta}{2n^2}$,

$$
|\hat{\mathbf{V}}_{ij}(t) - \mathbf{V}_{ij}(0)| \le \frac{8 R_v}{\alpha \sqrt{2\pi}} + \frac{8 R_v (m/\delta)^{1/d}}{\alpha} \le \frac{12 R_v (m/\delta)^{1/d}}{\alpha}.
$$

Taking a union bound, with probability $1 - \delta/2$,

$$
\|\hat{\mathbf{V}}(t) - \mathbf{V}(0)\|_F = \sqrt{\sum_{i,j} |\hat{\mathbf{V}}_{ij}(t) - \mathbf{V}_{ij}(0)|^2} \le \frac{12 n R_v (m/\delta)^{1/d}}{\alpha}.
$$

Bounding the spectral norm by the Frobenous norm,

$$\|\hat{\mathbf{V}}(t) - \mathbf{V}(0)\|_2 \leq \frac{12nR_v(m/\delta)^{1/d}}{\alpha}.$$

Taking $R_v = \frac{\alpha\lambda_0}{96n(m/\delta)^{1/d}}$ gives the desired bound.

$$\|\hat{\mathbf{V}}(t) - \mathbf{V}(0)\|_2 \leq \frac{\lambda_0}{8}.$$

$\square$

**Proof of Lemma B.4:**
To bound $\|\mathbf{V}(t) - \mathbf{V}(0)\|_2$ we now consider $\|\mathbf{V}(t) - \hat{\mathbf{V}}(t)\|_2$. The entries of $\mathbf{V}_{ij}(t)$ are given as

$$\mathbf{V}_{ij}(t) = \frac{1}{m}\sum_{k=1}^{m}\left\langle \mathbf{x}_i^{\mathbf{v}_k(t)^{\perp}},\ x_j^{\mathbf{v}_k(t)^{\perp}}\right\rangle \mathbb{1}_{ik}(t)\mathbb{1}_{jk}(t)\left(\frac{\alpha c_k \cdot g_k}{\|\mathbf{v}_k(0)\|_2}\right)^2.$$

The surrogate $\hat{\mathbf{V}}(t)$ is defined as

$$\hat{\mathbf{V}}_{ij}(t) = \frac{1}{m}\sum_{k=1}^{m}\left\langle \mathbf{x}_i^{\mathbf{v}_k(t)^{\perp}},\ x_j^{\mathbf{v}_k(t)^{\perp}}\right\rangle \mathbb{1}_{ik}(t)\mathbb{1}_{jk}(t).$$

The only difference is in the second layer terms. The difference between each entry is written as

$$|\mathbf{V}_{ij}(t) - \hat{\mathbf{V}}_{ij}(t)| = \left|\frac{1}{m}\sum_{k=1}^{m}\left\langle \mathbf{x}_i^{\mathbf{v}_k(t)^{\perp}},\ x_j^{\mathbf{v}_k(t)^{\perp}}\right\rangle \mathbb{1}_{ik}(t)\mathbb{1}_{jk}(t)\left(\left(\frac{\alpha c_k \cdot g_k}{\|\mathbf{v}_k(t)\|_2}\right)^2 - 1\right)\right|$$

$$\leq \max_{1\leq k\leq m}\left(\frac{\alpha^2 g_k(t)^2}{\|\mathbf{v}_k(t)\|_2^2} - 1\right).$$

Write $1 = \frac{\alpha^2 g_k^2(0)}{\|\mathbf{v}_k(0)\|_2^2}$, since $\|\mathbf{v}_k(t)\|_2$ is increasing in $t$ according to (2.3)

$$\frac{\alpha^2 g_k(t)^2}{\|\mathbf{v}_k(t)\|_2^2} - 1 = \frac{\alpha^2 g_k(t)^2}{\|\mathbf{v}_k(t)\|_2^2} - \frac{\alpha^2 g_k(0)^2}{\|\mathbf{v}_k(0)\|_2^2} \leq 3R_g(m/\delta)^{1/d} + 3R_v(m/\delta)^{1/d}/\alpha.$$

The above inequality is shown by considering different cases for the sign of the difference $g_k(t) - g_k(0)$. Now

$$\left|\frac{\alpha^2 g_k(t)^2}{\|\mathbf{v}_k(t)\|_2^2} - \frac{\alpha^2 g_k(0)^2}{\|\mathbf{v}_k(0)\|_2^2}\right| = \left|\left(\frac{\alpha g_k(t)}{\|\mathbf{v}_k(t)\|_2} + \frac{\alpha g_k(0)}{\|\mathbf{v}_k(0)\|_2}\right)\left(\frac{\alpha g_k(t)}{\|\mathbf{v}_k(t)\|_2} - \frac{\alpha g_k(0)}{\|\mathbf{v}_k(0)\|_2}\right)\right|$$

$$\leq \left|\left(\frac{\alpha g_k(0) + \alpha R_g}{\|\mathbf{v}_k(0)\|_2} + \frac{\alpha g_k(0)}{\|\mathbf{v}_k(0)\|_2}\right)\left(\frac{\alpha g_k(t)}{\|\mathbf{v}_k(t)\|_2} - \frac{\alpha g_k(0)}{\|\mathbf{v}_k(0)\|_2}\right)\right|$$

$$\leq (2 + R_g(m/\delta)^{1/d})\left|\left(\frac{\alpha g_k(t)}{\|\mathbf{v}_k(t)\|_2} - \frac{\alpha g_k(0)}{\|\mathbf{v}_k(0)\|_2}\right)\right|$$

$$\leq (2 + R_g(m/\delta)^{1/d})\max\left(\left|\frac{\alpha(g_k(0) + R_g)}{\|\mathbf{v}_k(0)\|_2} - \frac{\alpha g_k(0)}{\|\mathbf{v}_k(0)\|_2}\right|, \left|\frac{\alpha(g_k(0) - R_g)}{\|\mathbf{v}_k(0)\|_2 + R_v} - \frac{\alpha g_k(0)}{\|\mathbf{v}_k(0)\|_2}\right|\right)$$

$$\leq (2 + R_g(m/\delta)^{1/d})\max\left(R_g(m/\delta)^{1/d}, R_g(m/\delta)^{1/d} + R_v(m/\delta)^{1/d}/\alpha\right)$$

$$\leq 3R_g(m/\delta)^{1/d} + 3R_v(m/\delta)^{1/d}/\alpha,$$

where the second inequality holds due to Lemma B.10 with probability $1 - \delta$ over the initialization.

Hence:

$$\|\hat{\mathbf{V}}(t) - \mathbf{V}(t)\|_2 \leq \|\hat{\mathbf{V}}(t) - \mathbf{V}(t)\|_F = \sqrt{\sum_{i,j}|\hat{\mathbf{V}}_{ij}(t) - \mathbf{V}_{ij}(t)|^2} \leq 3nR_g(m/\delta)^{1/d} + 3nR_v(m/\delta)^{1/d}/\alpha.$$

Substituting $R_v, R_g$ gives

$$\|\hat{\mathbf{V}}(t) - \mathbf{V}(t)\|_2 \le \frac{\lambda_0}{8}.$$

Now we use Lemma B.3 to get that with probability $1 - \delta$

$$\|\hat{\mathbf{V}}(t) - \mathbf{V}(0)\|_2 \le \frac{\lambda_0}{8}$$

combining we get with probability $1 - \delta$

$$\|\mathbf{V}(t) - \mathbf{V}(0)\|_2 \le \frac{\lambda_0}{4}.$$

We note that the source for all the high probability uncertainty $1 - \delta$ all arise from initialization and the application of Lemma B.10. $\qquad\square$

**Proof of Lemma B.5:**
To prove the claim we consider each entry $i, j$ of $\mathbf{G}(t) - \mathbf{G}(0)$. We have,

$$|\mathbf{G}_{ij}(t) - \mathbf{G}_{ij}(0)| = \left| \frac{1}{m} \sum_{k=1}^{m} \sigma\left( \frac{\mathbf{v}_k(t)^\top \mathbf{x}_i}{\|\mathbf{v}_k(t)\|_2} \right) \sigma\left( \frac{\mathbf{v}_k(t)^\top \mathbf{x}_j}{\|\mathbf{v}_k(t)\|_2} \right) - \sigma\left( \frac{\mathbf{v}_k(0)^\top \mathbf{x}_i}{\|\mathbf{v}_k(0)\|_2} \right) \sigma\left( \frac{\mathbf{v}_k(0)^\top \mathbf{x}_j}{\|\mathbf{v}_k(0)\|_2} \right) \right|$$

$$\le \frac{1}{m} \left| \sum_{k=1}^{m} \sigma\left( \frac{\mathbf{v}_k(t)^\top \mathbf{x}_i}{\|\mathbf{v}_k(t)\|_2} \right) \sigma\left( \frac{\mathbf{v}_k(t)^\top \mathbf{x}_j}{\|\mathbf{v}_k(t)\|_2} \right) - \sigma\left( \frac{\mathbf{v}_k(t)^\top \mathbf{x}_i}{\|\mathbf{v}_k(t)\|_2} \right) \sigma\left( \frac{\mathbf{v}_k(0)^\top \mathbf{x}_j}{\|\mathbf{v}_k(0)\|_2} \right) \right|$$

$$+ \frac{1}{m} \left| \sum_{k=1}^{m} \sigma\left( \frac{\mathbf{v}_k(t)^\top \mathbf{x}_i}{\|\mathbf{v}_k(t)\|_2} \right) \sigma\left( \frac{\mathbf{v}_k(0)^\top \mathbf{x}_j}{\|\mathbf{v}_k(0)\|_2} \right) - \sigma\left( \frac{\mathbf{v}_k(0)^\top \mathbf{x}_i}{\|\mathbf{v}_k(0)\|_2} \right) \sigma\left( \frac{\mathbf{v}_k(0)^\top \mathbf{x}_j}{\|\mathbf{v}_k(0)\|_2} \right) \right|$$

$$\le 2 \left\| \frac{\mathbf{v}_k(t)}{\|\mathbf{v}_k(t)\|_2} - \frac{\mathbf{v}_k(0)}{\|\mathbf{v}_k(0)\|_2} \right\|_2 \le \frac{2\tilde{R}_v (m/\delta)^{1/d}}{\alpha}.$$

In the last inequality we used the fact,

$$\left\| \frac{\mathbf{v}_k(0)}{\|\mathbf{v}_k(0)\|_2} - \frac{\mathbf{v}_k(t)}{\|\mathbf{v}_k(t)\|_2} \right\|_2 \le \frac{\|\mathbf{v}_k(t) - \mathbf{v}_k(0)\|_2}{\|\mathbf{v}_k(0)\|_2} \le \frac{(m/\delta)^{1/d}}{\alpha} \|\mathbf{v}_k(t) - \mathbf{v}_k(0)\|_2,$$

where the first inequality uses that $\|\mathbf{v}_k(0)\|_2 \le \|\mathbf{v}_k(t)\|_2$ and is intuitive from a geometrical standpoint. Hence,

$$\|\mathbf{G}(t) - \mathbf{G}(0)\|_2 \le \|\mathbf{G}(t) - \mathbf{G}(0)\|_F = \sqrt{\sum_{i,j} |\mathbf{G}_{ij}(t) - \mathbf{G}_{ij}(0)|^2} \le \frac{2n\tilde{R}_v (m/\delta)^{1/d}}{\alpha\sqrt{2\pi}}.$$

Taking $\tilde{R}_v = \frac{\sqrt{2\pi}\alpha\mu_0}{8n(m/\delta)^{1/d}}$ gives the desired bound. Therefore, with probability $1 - \delta$,

$$\|\mathbf{G}(t) - \mathbf{G}(0)\|_2 \le \frac{\mu_0}{4}.$$

$\qquad\square$

Now that we have established bounds on $\mathbf{V}(t), \mathbf{G}(t)$ given that the parameters stay near initialization, we show that the evolution converges in that case:

**Proof of Lemma B.6:**
Consider the squared norm of the predictions $\|\mathbf{f}(t) - \mathbf{y}\|_2^2$. Taking the derivative of the loss with respect to time,

$$\frac{d}{dt} \|\mathbf{f}(t) - \mathbf{y}\|_2^2 = -2(\mathbf{f}(t) - \mathbf{y})^\top \left( \mathbf{G}(t) + \frac{\mathbf{V}(t)}{\alpha^2} \right) (\mathbf{f}(t) - \mathbf{y}).$$

Since we assume that $\lambda_{\min}\left(\mathbf{G}(t) + \frac{\mathbf{V}(t)}{\alpha^2}\right) \geq \frac{\omega}{2}$, the derivative of the squared norm is bounded as

$$\frac{d}{dt}\|\mathbf{f}(t) - \mathbf{y}\|_2^2 \leq -\omega\|\mathbf{f}(t) - \mathbf{y}\|_2^2.$$

Applying an integrating factor yields

$$\|\mathbf{f}(t) - \mathbf{y}\|_2^2 \exp(\omega t) \leq C.$$

Substituting the initial conditions, we get

$$\|\mathbf{f}(t) - \mathbf{y}\|_2^2 \leq \exp(-\omega t)\|\mathbf{f}(0) - \mathbf{y}\|_2^2.$$

$\square$

For now, assuming the linear convergence derived in Lemma B.6, we bound the distance of the parameters from initialization. Later we combine the bound on the parameters and Lemmas B.4, B.5 bounding the least eigenvalue of $\mathbf{\Lambda}(t)$, to derive a condition on the over-parameterization $m$ and ensure convergence from random initialization.

### Proof of Lemma B.7:

Denote $f(\mathbf{x}_i)$ at time $t$ as $f_i(t)$. Since $\|\mathbf{x}_i^{\mathbf{v}_k(t)^\perp}\|_2 \leq \|\mathbf{x}_i\|_2 = 1$, we have that

$$\left\|\frac{d\mathbf{v}_k(t)}{dt}\right\|_2 = \left\|\sum_{i=1}^n (y_i - f_i(t))\frac{1}{\sqrt{m}}c_k g_k(t)\frac{1}{\|\mathbf{v}_k(t)\|_2}\mathbf{x}_i^{\mathbf{v}^\perp}\mathbb{1}_{ik}(t)\right\|_2$$

$$\leq \frac{1}{\sqrt{m}}\sum_{i=1}^n |y_i - f_i(t)|\frac{c_k g_k(t)}{\|\mathbf{v}_k(t)\|_2}.$$

Now using (2.3) and the initialization $\|\mathbf{v}_k(0)\| = \alpha g_k(0)$, we bound $\left|\frac{c_k g_k(t)}{\|\mathbf{v}_k(t)\|_2}\right|$,

$$\left|\frac{c_k g_k(t)}{\|\mathbf{v}_k(t)\|_2}\right| \leq \left|c_k\left(\frac{g_k(0) + R_g}{\|\mathbf{v}_k(0)\|_2}\right)\right| \leq \frac{1}{\alpha}\left(1 + \alpha R_g/\|\mathbf{v}_k(0)\|_2\right).$$

By Lemma B.10, we have that with probability $1 - \delta$ over the initialization,

$$\alpha/\|\mathbf{v}_k(0)\|_2 \leq C(m/\delta)^{1/d}.$$

Hence $\alpha R_g/\|\mathbf{v}_k(0)\|_2 \leq 1$. This fact bounds $\left|\frac{c_k g_k(t)}{\|\mathbf{v}_k(t)\|_2}\right|$ with probability $1 - \delta$ for each $k$,

$$\left|\frac{c_k g_k(t)}{\|\mathbf{v}_k(t)\|_2}\right| \leq 2/\alpha.$$

Substituting the bound,

$$\left\|\frac{d}{dt}\mathbf{v}_k(t)\right\|_2 \leq \frac{2}{\alpha\sqrt{m}}\sum_{i=1}^n |f_i(t) - y_i|$$

$$\leq \frac{2\sqrt{n}}{\alpha\sqrt{m}}\|\mathbf{f}(t) - \mathbf{y}\|_2$$

$$\leq \frac{2\sqrt{n}}{\alpha\sqrt{m}}\exp(-\omega t/2)\|\mathbf{f}(0) - \mathbf{y}\|_2.$$

Thus, integrating and applying Jensen's inequality,

$$\|\mathbf{v}_k(t) - \mathbf{v}_k(0)\|_2 \leq \int_0^s \left\|\frac{d\mathbf{v}_k(s)}{dt}\right\|_2 ds \leq \frac{4\sqrt{n}\|\mathbf{f}(0) - \mathbf{y}\|_2}{\alpha\omega\sqrt{m}}.$$

Note that the condition $|g_k(t) - g_k(0)| \leq R_g$ is stronger than needed and merely assuring that $|g_k(t) - g_k(0)| \leq 1/(m/\delta)^{1/d}$ suffices. $\square$

Analogously we derive bounds for the distance of $g_k$ from initialization.

**Proof of Lemma B.8:**
Consider the magnitude of the derivative $\frac{dg_k}{dt}$,

$$\left|\frac{dg_k}{dt}\right| = \left|\frac{1}{\sqrt{m}}\sum_{j=1}^{n}(f_j - y_j)\frac{c_k}{\|\mathbf{v}_k\|_2}\sigma(\mathbf{v}_k^\top\mathbf{x}_j)\right|.$$

Note

$$\left|\frac{c_k}{\|\mathbf{v}_k\|_2}\sigma(\mathbf{v}_k^\top\mathbf{x}_j)\right| = \left|\sigma\left(\frac{\mathbf{v}_k^\top\mathbf{x}_j}{\|\mathbf{v}_k\|_2}\right)\right| \le 1$$

Thus applying Cauchy Schwartz

$$\left|\frac{dg_k(t)}{dt}\right| \le \frac{2\sqrt{n}}{\sqrt{m}}\|\mathbf{f}(t) - \mathbf{y}\|_2 \le \frac{2\sqrt{n}}{\sqrt{m}}\exp(-\omega t/2)\|\mathbf{f}(0) - \mathbf{y}\|_2,$$

and integrating from $0$ to $t$ yields

$$|g_k(t) - g_k(0)| \le \int_0^t\left|\frac{dg_k}{dt}(s)\right|ds \le \int_0^t\frac{2\sqrt{n}}{\sqrt{m}}\exp(-\omega s/2)\|\mathbf{f}(0) - \mathbf{y}\|_2 ds \le \frac{4\sqrt{n}\|\mathbf{y} - \mathbf{f}(0)\|_2}{\sqrt{m}\omega}.$$

$\square$

**Proof of Lemma B.9:**
Consider the $i$th entry of the network at initialization,

$$f_i(0) = \frac{1}{\sqrt{m}}\sum_{k=1}^{m}c_k\sigma\left(\frac{g_k\mathbf{v}_k^\top\mathbf{x}_i}{\|\mathbf{v}_k\|_2}\right).$$

Since the network is initialized randomly and $m$ is taken to be large we apply concentration to bound $f_i(0)$ for each $i$. Define $z_k = c_k\sigma\left(\frac{g_k(0)\mathbf{v}_k(0)^\top\mathbf{x}_i}{\|\mathbf{v}_k(0)\|_2}\right)$. Note that $z_k$ are independent sub-Gaussian random variables with

$$\|\mathbf{z}_k\|_\psi \le \|N(0,1)\|_\psi = C.$$

Here $\|\cdot\|_\psi$ denotes the 2-sub-Gaussian norm, (see [31] for example). Applying Hoeffding's inequality bounds $f_i(0)$ as

$$\mathbb{P}(|\sqrt{m}f_i(0)| > t) \le 2\exp\left(-\frac{t^2/2}{\sum_{k=1}^{m}\|\mathbf{z}_k\|_{\psi_2}}\right)$$

$$= 2\exp\left(\frac{-t^2}{2mC}\right).$$

Which gives with probability $1 - \delta/n$ that

$$|f_i(0)| \le \tilde{C}\sqrt{\log(n/\delta)}.$$

Now with probability $1 - \delta$ we have that, for each $i$,

$$|f_i(0) - y_i| \le |y_i| + \tilde{C}\sqrt{\log(n/\delta)} \le C_2\sqrt{\log(n/\delta)}.$$

Since $y_i = O(1)$. Hence, with probability $1 - \delta$,

$$\|\mathbf{f}(0) - \mathbf{y}\|_2 \le C\sqrt{n\log(n/\delta)}.$$

$\square$

**Proof of Lemma B.10:**
At initialization $\mathbf{v}_k \sim N(0, \alpha^2\mathbf{I})$ so the norm behaves like $\|\mathbf{v}_k(0)\|_2^2 \sim \alpha^2\chi_d$. The cumulative density of a chi-squared distribution with $d$ degrees of freedom behaves like $F(x) = \Theta(x^{d/2})$ for

small $x$ so we have that with probability $1 - \frac{\delta}{m}$, that $\|\mathbf{v}_k(0)\|_2 \geq \alpha(m/\delta)^{\frac{1}{d}}$ where $d$ is the input dimension. Applying a union bound, with probability $1 - \delta$, for all $1 \leq k \leq m$,

$$\frac{1}{\|\mathbf{v}_k(0)\|_2} \leq \frac{(m/\delta)^{1/d}}{\alpha}.$$

Now by (2.3) for $t \geq 0$, $\|\mathbf{v}_k(t)\|_2 \geq \|\mathbf{v}_k(0)\|_2$ so

$$\frac{1}{\|\mathbf{v}_k(t)\|_2} \leq \frac{1}{\|\mathbf{v}_k(0)\|_2} \leq \frac{(m/\delta)^{1/d}}{\alpha}.$$

$\square$

# E    PROOFS OF LEMMAS FROM APPENDIX C

**Proof of Lemma C.1:**
Fix $R$, without the loss of generality we write $S_i$ for $S_i(R)$. For each $k$, $\mathbf{v}_k(0)$ is initialized independently via $\sim N(0, \alpha^2 \mathbf{I})$, and for a given $k$, the event $\mathbb{1}_{ik}(0) \neq \mathbb{1}\{\mathbf{v}^\top \mathbf{x}_i \geq 0\}$ corresponds to $|\mathbf{v}_k(0)^\top \mathbf{x}_i| \leq R$. Since $\|\mathbf{x}_i\|_2 = 1$, $\mathbf{v}_k(0)^\top \mathbf{x}_i \sim N(0, \alpha^2)$. Denoting the event that an index $k \in S_i$ as $A_{i,k}$, we have

$$\mathbb{P}(A_{i,k}) \leq \frac{2R}{\alpha\sqrt{2\pi}}.$$

Next the cardinality of $S_i$ is written as

$$|S_i| = \sum_{k=1}^{m} \mathbb{1}_{A_{i,k}}.$$

Applying Lemma D.1, with probability $1 - \delta/n$,

$$|S_i| \leq \frac{2mR}{\alpha\sqrt{2\pi}} + \frac{16 \log(n/\delta)}{3}.$$

Taking a union bound, with probability $1 - \delta$, for all $i$ we have that

$$|S_i| \leq \frac{2mR}{\alpha\sqrt{2\pi}} + \frac{16 \log(n/\delta)}{3}.$$

$\square$

**Proof of Lemma C.2:**
To show this we bound the difference $g_k(s) - g_k(0)$ as the sum of the iteration updates. Each update is written as

$$\left| \frac{\partial L(s)}{\partial g_k} \right| = \left| \frac{1}{\sqrt{m}} \sum_{i=1}^{n} (f_i(s) - y_i) \frac{c_k}{\|\mathbf{v}_k(s)\|_2} \sigma(\mathbf{v}_k(s)^\top \mathbf{x}_i) \right|.$$

As $\left| c_k \sigma\left( \frac{\mathbf{v}_k(s)^\top \mathbf{x}_i}{\|\mathbf{v}_k(s)\|_2} \right) \right| \leq 1$,

$$\left| \frac{\partial L(s)}{\partial g_k} \right| \leq \frac{1}{\sqrt{m}} \sum_{i}^{n} |f_i(s) - y_i| \leq \frac{\sqrt{n}}{\sqrt{m}} \|\mathbf{f}(s) - \mathbf{y}\|_2.$$

By the assumption in the statement of the lemma,

$$\left| \frac{\partial L(s)}{\partial g_k} \right| \leq \frac{\sqrt{n}(1 - \frac{\eta\omega}{2})^{s/2} \|\mathbf{f}(0) - \mathbf{y}\|_2}{\sqrt{m}}.$$

Hence bounding the difference by the sum of the gradient updates:

$$|g_k(K+1) - g_k(0)| \le \eta \sum_{s=0}^{K} \left| \frac{\partial L(s)}{\partial g_k} \right| \le \frac{4\eta\sqrt{n}\|\mathbf{f}(0) - \mathbf{y}\|_2}{\sqrt{m}} \sum_{s=0}^{K} (1 - \frac{\eta\omega}{2})^{s/2}.$$

The last term yields a geometric series that is bounded as

$$\frac{1}{1 - \sqrt{1 - \frac{\eta\omega}{2}}} \le \frac{4}{\eta\omega},$$

Hence

$$|g_k(K+1) - g_k(0)| \le \frac{4\sqrt{n}\|\mathbf{f}(0) - \mathbf{y}\|_2}{\omega\sqrt{m}}.$$

$\square$

**Proof of Lemma C.3:**

To show this we write $\mathbf{v}_k(s)$ as the sum of gradient updates and the initial weight $\mathbf{v}_k(0)$. Consider the norm of the gradient of the loss with respect to $\mathbf{v}_k$,

$$\|\nabla_{\mathbf{v}_k} L(s)\|_2 = \left\| \frac{1}{\sqrt{m}} \sum_{i=1}^{n} (f_i(s) - y_i) \frac{c_k g_k(s)}{\|\mathbf{v}_k(s)\|_2} \mathbb{1}_{ik}(s) \mathbf{x}_i^{\mathbf{v}_k(s)^\perp} \right\|_2.$$

Since $\|\mathbf{v}_k(s)\|_2 \ge \|\mathbf{v}_k(0)\|_2 \ge \alpha(\delta/m)^{1/d}$ with probability $1 - \delta$ over the initialization, applying Cauchy Schwartz's inequality gives

$$\|\nabla_{\mathbf{v}_k} L(s)\|_2 \le \frac{(1 + R_g(m/\delta)^{1/d})\sqrt{n}\|\mathbf{f}(s) - \mathbf{y}\|_2}{\alpha\sqrt{m}}. \tag{E.1}$$

By the assumption on $\|\mathbf{f}(s) - \mathbf{y}\|_2$ this gives

$$\|\nabla_{\mathbf{v}_k} L(s)\|_2 \le \frac{2\sqrt{n}(1 - \frac{\eta\omega}{2})^{s/2}\|\mathbf{f}(0) - \mathbf{y}\|_2}{\alpha\sqrt{m}}.$$

Hence bounding the parameter trajectory by the sum of the gradient updates:

$$\|\mathbf{v}_k(K+1) - \mathbf{v}_k(0)\|_2 \le \eta \sum_{s=0}^{K} \|\nabla_{\mathbf{v}_k} L(s)\|_2 \le \frac{2\sqrt{n}\|\mathbf{f}(0) - \mathbf{y}\|_2}{\alpha\sqrt{m}} \sum_{s=1}^{K} \left( 1 - \frac{\eta\omega}{2} \right)^{s/2}$$

yields a geometric series. Now the series is bounded as

$$\frac{1}{1 - \sqrt{1 - \frac{\eta\omega}{2}}} \le \frac{4}{\eta\omega},$$

which gives

$$\|\mathbf{v}_k(K+1) - \mathbf{v}_k(0)\|_2 \le \frac{8\sqrt{n}\|\mathbf{f}(0) - \mathbf{y}\|_2}{\alpha\sqrt{m}\omega}.$$

$\square$

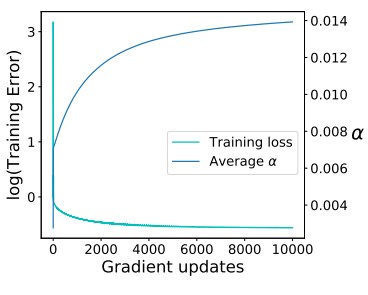 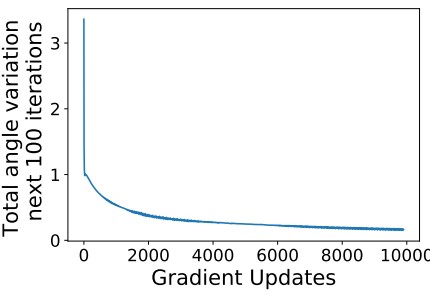

(a) Average $\alpha$ during training

(b) Total variation in angle, next 100 steps

Figure 2: Training dynamics of networks from class (1.2) on digits 4,9 of MNIST

## F  NUMERICAL ILLUSTRATION

Below we provide preliminary numerical illustrations demonstrating the phase transition between **V**-dominated to **G**-dominated convergence. We follow the settings introduced in Section 2. In the experiments we run WeightNorm gradient descent for $K = 10000$ gradient iterations and learning rate $\eta = 3 \times 10^{-3}$. We train the neural network from class (1.2), with $m = 10000, n = 1000, d = 784, \alpha = 10^{-3}$ and use the first 1000 images of the digits 4 and 9 of the MNIST dataset. We restrict to the digits of 4 and 9 since we focus on the scalar output network with two classes.

To compute $\alpha$ for the network during training we calculate the ratio,

$$\alpha(s) = \frac{\sum_{k=1}^{m} \|\mathbf{v}_k(s)\|_2}{\sum_{k=1}^{m} g_k(s)}.$$

During training we note that $\alpha$ grows, as the training loss decreases and the convergence rate slows down (emergence of the **G**-dominated regime). To measure the change in the direction of $\mathbf{v}_k(t)$ during training we compute the angle variation of the parameter, this is computed for each neuron for each gradient update as,

$$\Delta\theta_k(s) = \arccos\left(\frac{\left|\langle \mathbf{v}_k(s), \mathbf{v}_k(s+1) \rangle\right|}{\|\mathbf{v}_k(s)\|_2 \|\mathbf{v}_k(s+1)\|_2}\right).$$

To visualize the angle change of the network during training, we compute the mean angle change for each iteration, $\Delta_{\text{ave}}(s) = \frac{1}{m} \sum_{k=1}^{m} \Delta\theta_k(s)$. We plot a moving sum of $\Delta_{\text{ave}}(s)$ window of size 100, $\Delta\theta_{\text{window}}(s) = \sum_{r=s}^{s+100} \Delta_{\text{ave}}(r)$ to measure the change of the angle throughout training.

