# OpenReview forum: "On the Dynamics and Convergence of Weight Normalization for Training Neural Networks"
_ICLR.cc/2020/Conference — Reject_

### Official Review · AnonReviewer3 · 2019-10-22
**Official Blind Review #3**

**Rating:** 3

**Review:**

Global convergence of NNs is an important research direction in deep learning. There have been significant progresses in this direction since last year. Most noticeable, the Neural tangent kernels (NTK) [1], which shows in the infinite width setting, NTK is deterministic and remains almost constant during gradient descent. NNs are essentially the same as kernel methods. Proofs of global convergence of NNs (without normalization) are built on this intuition.

This is the first paper (to my best knowledge) to prove such convergence result for weight normalization. The paper is very well written and the presentation is very clear and I really enjoy reading it! The proof builds on (and very similar to) [2] and other recent works concerning the global convergence of NNs in the kernel regime. The main strategy is as follows:
step 1. show the finite-width NTK concentrates near the infinite width NTK, whose least eigenvalue is positive (need proof)
step 2. show the change of finite-width NTK is tiny and essentially does not alter the least eigenvalue.

As such, the optimization problem is essentially a strong convex problem. Although the high level picture is very similar to previous convergence papers, the technical details are different. In particular, the authors show some improvements over  previous works: e.g. the width required for the proof here is n^4 (n == dataset size) rather than n^6 without normalization ([3]). I think these bounds are very loose and it is not clear to me if normalization provably reduces the degree of over-parameterization; see [4].  There are also some other interesting results concerning weight normalization, e.g. the NTK could be decoupled into the sume of the `directional`   NTK and the `length`  NTK.

Overall, this is a good paper but the contribution might not be sufficient for acceptance for the reasons below.
1. the overall framework is very similar to previous works and the dynamics of NNs still lives in the kernel regime, which might not be the most interesting regime for deep learning.
2. In this kernel regime, global convergence (or GD dynamics) might not be the most interesting object to study since we already have many existing works in this direction. I hope to see new insights beyond global convergence, e.g. benefits of normalization to generalization (potentially for multilayer networks), etc. See comments below.


Comments:
1. Extending to multilayer?
2. Show normalization provably reduces the level of over-parameterization needed for convergence. This requires a lower bound for the width that NN cannot converge if the width is below the threshold.
3. What can we say about generalization: normalization vs no normalization? e.g. [5]





[1] Arthur Jacot, Franck Gabriel, and Clément Hongler. Neural tangent kernel: Convergence and generalization in
neural networks. In Advances in neural information processing systems, pages 8571–8580, 2018.
[2] Simon S. Du, Xiyu Zhai, Barnabas Poczos, and Aarti Singh. Gradient descent provably optimizes overparameterized neural networks.
[3]Sanjeev Arora, Simon Du, Wei Hu, Zhiyuan Li, and Ruosong Wang. Fine-grained analysis of optimization
and generalization for overparameterized two-layer neural networks. In International Conference on Machine
Learning, pages 322–332, 2019.
[4]Samet Oymak, Mahdi Soltanolkotabi. Towards moderate overparameterization: global convergence guarantees for training shallow neural networks
[5] Sanjeev Arora, Simon Du, Wei Hu, Zhiyuan Li, and Ruosong Wang. Fine-grained analysis of optimization
and generalization for overparameterized two-layer neural networks. In International Conference on Machine
Learning, pages 322–332, 2019.


**Experience Assessment:**

I have published one or two papers in this area.

**Review Assessment: Checking Correctness Of Derivations And Theory:**

I assessed the sensibility of the derivations and theory.

**Review Assessment: Checking Correctness Of Experiments:**

N/A

**Review Assessment: Thoroughness In Paper Reading:**

I read the paper at least twice and used my best judgement in assessing the paper.

---

> ### Author Response · Authors · 2019-11-09
> **Response to Reviewer #3**
>
> Thank you for taking the time to review our work. We are glad to hear that you enjoyed reading it!
>
> As mentioned in our introduction, we have analyzed a different model from the one in [1] and like you said, while the general techniques in our proof are the same (and are shared among the many recent works of the NTK, such as [2,3,5,6] to name a few), we uncover previously unexplored dynamics of the NTK for normalization methods and specifically WeightNorm. Under our formulation, we find for the first time that normalization methods split the NTK into parts. Of additional interest is that unlike previous works that investigated the un-normalized settings, the new NTKs that emerge satisfy different scaling dynamics (e.g., over-parameterization that depends on alpha and scale independence as it appears in the G dominated convergence).
>
> Regarding [4] (Oymak et al.). Thank you for pointing out this reference. We were not aware of this result. Compared to our results, the overparameterization is actually weaker than the G-dominated regime if no assumptions are made about the data. Only the corollaries presented in [4], which make additional assumptions on the data distribution (namely that the data is coming from a high dimensional Gaussian), improve the over-parameterization compared with our results which hold for general data. We have now cited this interesting work in the revision.
>
> We thank the reviewer for suggesting insightful questions that we have not addressed in this work, our outlook on the comments suggested are:
>
> Addressing comment 1 regarding multilayer analysis: While we believe that it is possible to extend our convergence argument to multilayer networks, the NTK decomposition occurs in each layer so we found it most canonical to investigate the different basic properties of the length direction decomposition for a shallow neural network where more can be derived and the theory is more illuminating. Nonetheless, we believe that we can build a convergence proof for deep dynamically normalized networks extending the techniques presented in [3], [5], or [6] that study deep global convergence. Since one of the main focus of this work was in examining the different dynamics that occur in the different regimes of WeightNorm as compared with unnormalized training, we have not investigated such extensions in the present work.
>
> Addressing comment 2 regarding provable and necessary level of over-parameterization: This is a very interesting inquiry but we believe that current theory and the settings in which the NTK is studied, are not sufficient to obtain such lower bound results.
>
> Addressing comment 3 regarding generalization: Our most immediate next goal is to further investigate the produced kernels from the point of view of the regularization of the trajectory path and the resulting generalization. We believe that using our new NTK decomposition opens many interesting directions for analysis in that regard as we can study the spectral properties of the modified kernels during training.
>
> [1] Gradient Descent Provably Optimizes Over-parameterized Neural Networks
> Simon S. Du*, Xiyu Zhai*, Barnabás Póczos, Aarti Singh. (ICLR) 2019
>
> [2] Fine-Grained Analysis of Optimization and Generalization for Overparameterized Two-Layer Neural Networks, in Proc. of International Conference of Machine Learning (ICML) 2019
>
> [3] Generalization Bounds of Stochastic Gradient Descent for Wide and Deep Neural Networks
> Yuan Cao and Quanquan Gu, in Proc. of Advances in Neural Information Processing Systems (NeurIPS) 2019
>
> [4] Samet Oymak, Mahdi Soltanolkotabi. Towards moderate overparameterization: global convergence guarantees for training shallow neural networks
>
> [5] Algorithm-dependent Generalization Bounds for Overparameterized Deep Residual Networks, Spencer Frei, Yuan Cao and Quanquan Gu  in Proc. of Advances in Neural Information Processing Systems (NeurIPS) 2019
>
> [6] Gradient Descent Finds Global Minima of Deep Neural Networks
> Simon S. Du, Jason D. Lee, Haochuan Li, Liwei Wang, Xiyu Zhai*
> in Proc. of International Conference on Machine Learning (ICML) 2019.

---

### Official Review · AnonReviewer1 · 2019-10-22
**Official Blind Review #1**

**Rating:** 3

**Review:**

*Summary of the contributions:*
This paper deals with convergence, of the hidden layer of a 2-layers Relu Network trained with Weight Normalization which consists in decoupling the direction and the magnitude for the pre-activation layers. The authors show, under mild assumption, the linear convergence (with high probability) of the training loss in the over-parameterized setting.

Conceptually the result says that using random feature selection (the last layer is fixed, initialized with Rademacher random variables), with a large enough network, the hidden weights do not need to move much from their initialization to get arbitrarily small training loss.
Thus, the analysis of the whole dynamics is reduced to a perturbation analysis of the dynamics around the initialization. (I would like to emphasize that even though the final argument ends up to be a ‘simple’ strong convexity convergence proof, I acknowledge that the proof of such result is far from being straightforward.)

The contributions of this paper are the following: in the setting introduced in [15] (the ref numbers correspond to the ones in the paper) the authors prove the linear convergence of the training loss for a large enough output layer of a 2-layer Relu network with Weight Normalization. Regarding to [15] the difference is that the authors study Weight Normalization. They are able to get better bounds on the over parametrization, and provide an analysis that highlights two training regimes: a fast one (V-dominated) and a stable one (G-dominated) that could explain the benefit of weight normalization.


*Decision:*
I am leaning for a weak reject for the following reasons:
1. The main issue to me is that this work builds up on the work by [15] without addressing the issues risen by [12]. One important takeaway from [12] that the ‘lazy training’ regime studied might not be a desired behavior and might not happen in practice. Thus, I think that extending the work form [15] to WeightNormalization would be interesting if the phenomenons described by the theory are backed-up by some practical evidence.
For instance : “we believe that G-dominated convergence actually is common but that it emerges at later stages in training, after the weights have adopted their directions in the V-dominated regime and improves stability.“ is something that the authors could check in practice.

2. I found the discussion regarding the phase transition a bit unclear. The magnitude of the weights v_k is treated as it was alpha the coefficient used to parametrized their initialization. Then since the phase transition depends on the magnitude of \alpha, it is argued that the model naturally transitions as \|v_k\| increases (which is the case because of the discrete gradient updates).
However, to me to make that phase transition argument you would need to show that the scaling of the matrices at time actually depends on the norm of the current iterate v_k(t)

3. Finally, regarding the proof of Theorem 4.1 and 4.2, I found it way more clear in [12]. There is a key subtle bootstrap argument that is not explicitly stated in the proof of Theorem B.1 and B.2: The condition R’ < R  (stated in Lemma 3.4 in [12]) is the key argument that leads to a sufficient bound on the overparametrization.
I put some remark regarding the clarity of the proof in the minor comment section. I think it would improve the quality of the work but did not have a major impact on my grade.

4. The presentation is slightly misleading. The fact that the last layer is not optimized is slightly mentioned once and the author make it appear just like is was done without any loss of generality  “We consider the case where the ck’s are fixed. This does not alter the capacity of the network since the gks are trainable (ck and gk are interchangeable analytically).”. Moreover, the omission that Theorem 4.1 and 4.2 are only true with high probability makes even that fact even more unclear. I think that this aspect (last layer is fixed) should be clearly emphasized in the revision.



*Questions:*
“we believe that G-dominated convergence is common”, can you provides evidence that G-dominated and V-dominated  stages occurs the way you describe it in the discussion ?
Same for the claim “The direction of the weights changes rapidly at the earlier stages of training when α is small, and G-dominated convergence ensues as α grows, leading to improved stability.” Particularly the link between your theory, the norm of v_k and alpha. (c.f point 2. of the decision section)
I think that it is important to emphasize that in your work c_k are randomly sampled and then fixed (it is much more clear in [15], but I do not think that the reader should know the related work to realize that). You seem to argue that it is without loss of generality when you say “We consider the case where the ck’s are fixed. This does not alter the capacity of the network since the gks are trainable (ck and gk are interchangeable analytically).”
but this statement seems wrong since there is a relu activation between c_k and g_k (thus you cannot interchange negative c_k with negative g_k). Can you comment this ?
Can you clarify the bootstrap argument mentioned in section 3. of the decision section ?  More precisely, can you justify the sentence “this contradicts one of Lemmas B.7, B.8 at time T_0.” in the proof of Theorem B.1 ?



*Minor comments: (do not need to be addressed in the rebuttal)*
- In the proof of Theorems B.1 and B.2 the statement “clearly T_0>0” should justified. This statement is true but a continuity argument that should be stated. However, the dependence in t could be easily avoided using the proof technique from [12] (Lemma 3.2) where the result is actually stated for any weight close enough to the initialization.
- The title of [12] has changed and one author is now missing. (since you cite the NeurIPS version you should then cite the right title with all the authors)
- I would not say that  “smaller alpha leads to faster convergence” since the optimal step size depend on \alpha.


=== After rebuttal ===
I would like to thank the author for their detailed answer.
I still think that experimental evidence that the phenomenon described by the theory actually happens in practice should be provided. I maintain my score.


**Experience Assessment:**

I have published one or two papers in this area.

**Review Assessment: Checking Correctness Of Derivations And Theory:**

I assessed the sensibility of the derivations and theory.

**Review Assessment: Checking Correctness Of Experiments:**

N/A

**Review Assessment: Thoroughness In Paper Reading:**

N/A

---

> ### Author Response · Authors · 2019-11-09
> **Response to Reviewer #1 (part 1)**
>
> Thank you for taking the time to review our paper and for your thoughtful comments.
>
> Addressing point 1:
> The analysis we present is based on the NTK, which is an increasingly important technique in theoretical deep learning and and is also receiving significant interest from the broader ML community. In our work, we are investigating how normalization methods modify the NTK. Our analysis points out to striking new dynamics (namely the V and G regimes that exhibit different behavior) and provides proof to the non-linear ReLU networks that emerge in WeightNorm training. While [12] brought up very good points in regard to the value of the NTK in practice, this area of research is still the focus of much discussion. The extent to which the NTK can explain the successes of deep learning is still unclear. For example, [1] recently illustrated competitive performance on CIFAR10 using the theory of NTK and the 'kernel regime'. Overall, despite the strong arguments presented in [12], the study and analysis of the NTK and over-parameterized networks is still of great interest.
>
> Even so, our paper does address [12]. We note a new phenomena that is reminiscent of the scaling dynamics of lazy training. This occurs upon analyzing the G-dominated regime. Just like in lazy training, larger alpha or tau leads to more stable weights and we argue for a transition from a rapid initial stage (V-dominated) into a later more stable stage (G-dominated) which is similar to the regime in the work of [12]. This transition is supported by equation (2.3). Further, G-dominated convergence presents itself naturally from our analysis, without additional rescaling of the output of the network. The type of convergence we see when analyzing G is independent of alpha, which is not the case for un-normalized networks and is not trivial. In addition, in our work the convergence is illustrated in the non-smooth non-convex ReLU regime.
>
> Addressing point 3:
> We note our main results are Theorems 4.1 and 4.2, whose proofs are different from those of Theorems B.1 and B.2. We have expanded the explanations in Theorems B.1, B.2 to make the arguments clearer, especially in regard to the subtle bootstrap argument.
>
> The argument for convergence for Theorems B.1, B.2 is a bit tricky. As mentioned, is similar to the one in [15]. However, it needs to incorporate the two different regimes, which require different bounds and supporting Lemmas for each case. We are also sketching an outline of the general idea of the proofs here.
>
> Given the width m from the conditions of the theorem, we have that at initialization the least eigenvalue is bounded below (since Lemmas B.1, B.2 hold).
> Since the parameters of the model change continuously in gradient flow, we may take T_0 to be the first time point when the results of Lemmas B.7/ B.8 do not hold (this is when the eigenvalues are not bounded below or when the trajectory is too large). This T_0 is guaranteed to be positive since we may take arbitrarily small positive time for which the change in the weights is small enough so that Lemmas B.3, B.4, B.5 all hold, which would imply that all of the hypothesis of Lemmas B.7 B.8 hold at that small time.
>
> So we have that Lemmas B.7, B.8 hold for at least some interval 0< t < T_0, with T_0 being the first time one of the hypothesis of B.7, B.8 breaks. Nonetheless, since B.7, B.8 hold at times 0 < t < T_0, we have that the distance of the parameters from their initialization at time T_0 is sufficiently bounded by the respective R_v, R_g (We use the overparameterization and also apply Lemma B.9). With this bounded trajectory the hypothesis of Lemmas B.3, B.4, B.5 are all satisfied which implies that the hypothesis of Lemmas B.7 and B.8 are satisfied too at time T_0. Therefore we reach a contradiction based on our assumption that T_0 is the first failure point. With this we conclude by contradiction that the statements of Lemmas B.3, B.4, B.5 hold for all t >0, hence by Lemma B.6 we get our desired convergence for all t > 0 (which is different for each regime).

---

> > ### Author Response · Authors · 2019-11-09
> > **Response to Reviewer #1 (part 2)**
> >
> > Addressing point 4 and question 2:
> > In the over-parameterized regime and in our analysis, the training of c_k and g_k indeed turns out to be interchangeable. However, the reviewer is correct in pointing out that the reason for this is a bit deeper. The reason is that c_k are initialized from a Rademacher distribution and are bounded away from 0. Over-parameterization (even quite mild) ensures that the c_k’s do not change their sign during training, (i.e., showing that |c_k(t) -c_k(0)| < 1 for all t). With this fact, we can write c_k in terms of sign(c_k) and |c_k|. Computing the flow equations for g_k and |c_k| turns out to be of the same form, and we decided to just optimize g_k due to the shared dynamics. This analysis also points to c_k and g_k taking the same form of NTK and so analyzing the training of both layers turns out to be similar.
> > We rewrote the sentence discussing c_k and g_k in the revision to emphasize that during training we do not optimize the parameters c_k. Also to avoid any confusions regarding the interchangeability of c_k and g_k, we have modified this part of the sentence as well.
> > We believe that the setting where both c_k and g_k are trained simultaneously can be treated departing from the theory that we have presented. However, this will add additional technical details, that we leave for future work.
> > Overall, introducing the training of c_k will lead to the introduction of additional trainable parameters and would involve more Lemmas analogous to Lemmas B.2, B.5, and B.8.
> >
> > Thank you also for pointing out the missing 'high probability' clause, omitting which is indeed misleading. We have added this to the statement of the theorems and the statement that the results are with high probability 1- delta over the initialization.
> >
> > Addressing question 1 and point 2:
> > Verifying the hypothesis that G-dominated convergence is common will require a thorough experimental evaluation, which is not the intention of this article. Our reasoning for this idea is as follows. In practice it is observed that dynamical normalization methods allow one to use larger step sizes while maintaining good convergence. Now, for large step sizes, the scale of the weights will grow significantly, as illustrated in Figure 1, hence lead to a very small V term, as seen from eqs (3.5) and (3.3). We are also adding a numerical illustration in the Appendix. The added figure in the Appendix shows that in the small experiment conducted, as training proceeds, the training loss decreases and alpha grows, while the change in parameter direction slows.
> >
> > minor comments:
> > - fixed the citation of [12] in the new version
> >
> > - “smaller alpha leads to faster convergence”: That is correct in the case of gradient flow, since the step size is infinitesimal it does lead to faster convergence since there is no step-size. In the finite step case, we mention that this is balanced by small step size so the overall rate does not change. We have rephrased the sentences adding the word "APPARENT faster convergence".

---

### Official Review · AnonReviewer2 · 2019-10-23
**Official Blind Review #2**

**Rating:** 6

**Review:**

This paper presents a general proof of the convergence of two-layer ReLU networks with weight normalization trained with gradient descent. Weight normalization re-parameterizes the weights to decouple the directions and lengths of kernels. Depending on the lengths of kernels the training process can be divided into two regimes, corresponding to updates of lengths and directions, respectively. One of the regimes naturally corresponds to lazy training where the directions remain stable. And there are transitions from one regime to the other when the lengths gradually change during the training process.

The proofs are very solid and the results look strong to me.

My only concern is that this paper is obviously squeezed to fit into 8 pages. I am not sure whether this is acceptable. Personally I think there is no need to do so and it's okay to just use 10 pages based on the loaded contents of this paper.

=====================
I'd like to keep my original score for this paper after reading the authors' response.

**Experience Assessment:**

I have read many papers in this area.

**Review Assessment: Checking Correctness Of Derivations And Theory:**

I assessed the sensibility of the derivations and theory.

**Review Assessment: Checking Correctness Of Experiments:**

N/A

**Review Assessment: Thoroughness In Paper Reading:**

I read the paper at least twice and used my best judgement in assessing the paper.

---

> ### Author Response · Authors · 2019-11-09
> **Response to Review #2**
>
> Thank you for taking the time to review our paper and for your positive feedback.
>
> In regard to the formatting, we realized that our margins were formatted incorrectly. Thank you for pointing this out. The updated paper has 9 pages.

---

### Decision · Program_Chairs · 2019-12-19

**Decision:**

Reject

**Comment:**

The goal of this paper is to study the dynamics of convergence of neural network training when weight normalization is used. This is an important and interesting area. The authors focus on analyzing such effect based on a recent theoretical trend which studies neural network dynamics based on the so called neural tangent kernel (NTK). The authors show an interesting phenomena of length-direction decoupling. The reviewers raise various points some of which have been addressed by the authors in their response. Two main points not yet clearly addressed is (1) what is the novelty of the theoretical framework given existing literature and (2) what are the benefits of weight normalization based on this theory (e.g. generalization etc. ). The authors suggest improved convergence rate and overparameterization dependence (i.e. that with weight normalization the required width is decreased) as a possible advantage. However, as pointed out by reviewer 3 there are existing results which already obtain better results without weight normalization (the authors' response that this is only true in randomized scenarios is actually not accurate). Based on above I do not think the paper is ready for publication. That said I think this is a nice direction and well-written paper. I recommend the authors revise and resubmit to a future venue. Some suggestions for improvements in case this is helpful (1) improve literature review and discussion of existing results (2) identify clear benefits to weight normalization. I doubt that improving overparameterization in existing form is one of them unless you provide a lower-bound (I suspect one can eventually obtain even linear overparameterization i.e. number of parameters proportional to number of training data even in the NTK regime without weight normalization. The suggestion by the reviewer at looking at generalization might be a good direction to pursue.